# Mapping multimorbidity progression among 190 diseases
Shasha Han [1,2,3] ✉, Sairan Li[1], Yunhaonan Yang[4], Lihong Liu[5], Libing Ma[6], Zhiwei Leng[7],
Frances S. Mair [8], Christopher R. Butler [9,10], Bruno Pereira Nunes[11,12], J. Jaime Miranda [13,14],
Weizhong Yang[1,2,3], Ruitai Shao[1,2,3] & Chen Wang[1,2,3,15] ✉

## Abstract

**Background** Current clustering of multimorbidity based on the frequency of common disease combinations is inadequate. We estimated the causal relationships among prevalent diseases and mapped out the clusters of multimorbidity progression among them. **Methods** In this cohort study, we examined the progression of multimorbidity among 190 diseases among over 500,000 UK Biobank participants over 12.7 years of follow-up. Using a machine learning method for causal inference, we analyzed patterns of how diseases influenced and were influenced by others in females and males. We used clustering analysis and visualization algorithms to identify multimorbidity progress constellations. **Results** We show the top influential and influenced diseases largely overlap between sexes in chronic diseases, with sex-specific ones tending to be acute diseases. Patterns of diseases that influence and are influenced by other diseases also emerged (clustering significance $P_{au} > 0.87$), with the top influential diseases affecting many clusters and the top influenced diseases concentrating on a few, suggesting that complex mechanisms are at play for the diseases that increase the development of other diseases while share underlying causes exist among the diseases whose development are increased by others. Bi-directional multimorbidity progress presents substantial clustering tendencies both within and across International Classification Disease chapters, compared to uni-directional ones, which can inform future studies for developing cross-specialty strategies for multimorbidity. Finally, we identify 10 multimorbidity progress constellations for females and 9 for males (clustering stability, adjusted Rand index >0.75), showing interesting differences between sexes.

**Conclusion** Our findings could inform the future development of targeted interventions and provide an essential foundation for future studies seeking to improve the prevention and management of multimorbidity.

## Plain language summary

Mapping out clusters of diseases is crucial to addressing the rising challenge of co-occurrence of multiple diseases, known as multimorbidity. However, the current way of grouping diseases based on their associations isn't enough to understand how they develop over time. We've come up with a new approach to map out how groups of diseases progress together based on the strength of their causal relationships. By looking at how each disease affects the development of others, we can get a better understanding of how they form clusters. Our research goes beyond just showing which diseases occur together, and it's a step toward improving how we prevent and manage multiple health conditions in the future.

[1]School of Population Medicine and Public Health, Chinese Academy of Medical Sciences & Peking Union Medical College, Beijing, China. [2]State Key Laboratory of Respiratory Health and Multimorbidity, Beijing, China. [3]Key Laboratory of Pathogen Infection Prevention and Control (Peking Union Medical College), Ministry of Education, Beijing, China. [4]Section of Epidemiology and Population Health, West China Second University Hospital, Sichuan University, Chengdu, China. [5]China-Japan Friendship Hospital, Beijing, China. [6]Affiliated Hospital of Guilin Medical University, Guangxi, China. [7]Peking Union Hospital, Beijing, China. [8]School of Health and Wellbeing, College of Medicine, Veterinary and Life Sciences, University of Glasgow, Glasgow, UK. [9]Department of Brain Sciences, Imperial College London, London, UK. [10]Imperial College Healthcare NHS Trust, London, UK. [11]Postgraduate Program of Nursing, Federal University of Pelotas, Pelotas, Brazil. [12]Postgraduate Program of Epidemiology, Federal University of Pelotas, Pelotas, Brazil. [13]Sydney School of Public Health, Faculty of Medicine and Health, University of Sydney, Sydney, NSW, Australia. [14]CRONICAS Centre of Excellence in Chronic Diseases, Universidad Peruana Cayetano Heredia, Lima, Peru. [15]Chinese Academy of Medical Sciences & Peking Union Medical College, Beijing, China. ✉e-mail: hanshasha@pumc.edu.cn; wangchen@pumc.edu.cn

Multimorbidity, the co-occurrence of two or more long term conditions, is a rising health challenge worldwide[1,2]. It is commonly associated with the occurrence of geriatric syndromes, also being influenced by processes related to aging[3–6]. With the global population aging and the remarkable negative impact of chronic diseases, there is a growing need to understand how multimorbidity accumulates in individuals over time[7,8]. This understanding has the potential to help researchers better understand the mechanisms underpinning the development of multimorbidity, assist healthcare providers in intervening or screening for additional conditions when the first disease becomes noticeable, and enable decision-makers to develop approaches for integrated care management and better equip health services to meet the person-centered health needs[9–11].

When providing integrated and person-centered care to individuals with unmet health needs, it is crucial to understand how multimorbidity progresses[7]. One common way to achieve this is by identifying shared components among temporal sequences of diagnoses[12,13]. However, this method tends to overemphasize the most prevalent diagnoses, leading to large clusters where too many diseases from less relevant paths are grouped together. For instance, hypertension, a shared component by most temporal sequences[14], often dominates clustering and may result in the majority of diseases in one cluster. More importantly, researchers have flagged that these association-based clustering may be misleading in determining the trajectories of multimorbidity progression[7,11,15]. Diseases may be associated simply because they are commonly occurring, rather than being causally linked through biological interaction or underlying shared mechanisms. Additionally, the method does not allow bi-directional trajectories between diseases, where one disease can lead to another disease and vice versa, which could prevent reversed trajectories of progression. A more informative approach is to take into account bi-directional progress when evaluating patterns of multimorbidity progress[6,7].

We address this gap by providing a framework to examine the progression of multimorbidity and map the clustering of progression based on causal relationships between diseases. This approach entails a series of analyses, starting with examining how having one disease affects the development of another (i.e., one-step multimorbidity progression), for each directional pair of diseases. We then identify the top influential and influenced diseases, investigate shared mechanisms across diseases that drive the development of multimorbidity, and characterize multimorbidity

development pathways across current disease classifications. Finally, we cluster one-step progress into progress constellations or clusters. Using this framework, we analyzed the progression of multimorbidity among 190 diseases among over 500,000 UK Biobank participants and developed tools to visualize all the results in detail.

## Methods

### Study design and populations
This cohort study was carried out on 502,413 adult participants from UK Biobank, aged 37–73 years (mean [SD] age: 57.1 [8.1] years), enrolled from March 2006 to December 2010 with longitudinal follow-up (mean [SD], 12.7 [0.9] years). This study was covered by the ethical approval from the UKB granted by the National Information Governance Board for Health and Social Care and the NHS North West Multicenter Research Ethics Committee. All participants provided informed consent through electronic signature at baseline assessment. Ethical approval of the study was obtained from the Chinese Academy of Medical Sciences & Peking Union Medical College. All data extracted were deidentified for analysis.

### Disease status and baseline covariates
We focused on the diseases with prevalence rates greater than 1% in each sex, coded in the International Classification of Diseases (ICD-10) terminology, which identified a total of 190 most prevalent diseases (154 diseases in females and 160 diseases in males, Supplementary Methods). Sociodemographic factors, health behaviors, health status, all disease status and factors influencing health status and contact with health services, and family history were included in baseline covariates to control for measured confounding. Additionally, all disease status and factors influencing health status and contact with health services (ICD codes, Z00–Z99) at baseline were included to control confounding. These variables indicate a reason for an encounter or provide additional information about a patient encounter.

### Statistical analysis
Our framework for analysis is illustrated in Fig. 1. We used the target maximum likelihood estimation (TMLE) method to estimate the effect of developing one disease on the development of another condition[16]. TMLE is a doubly robust estimator, allowing for unbiased inference

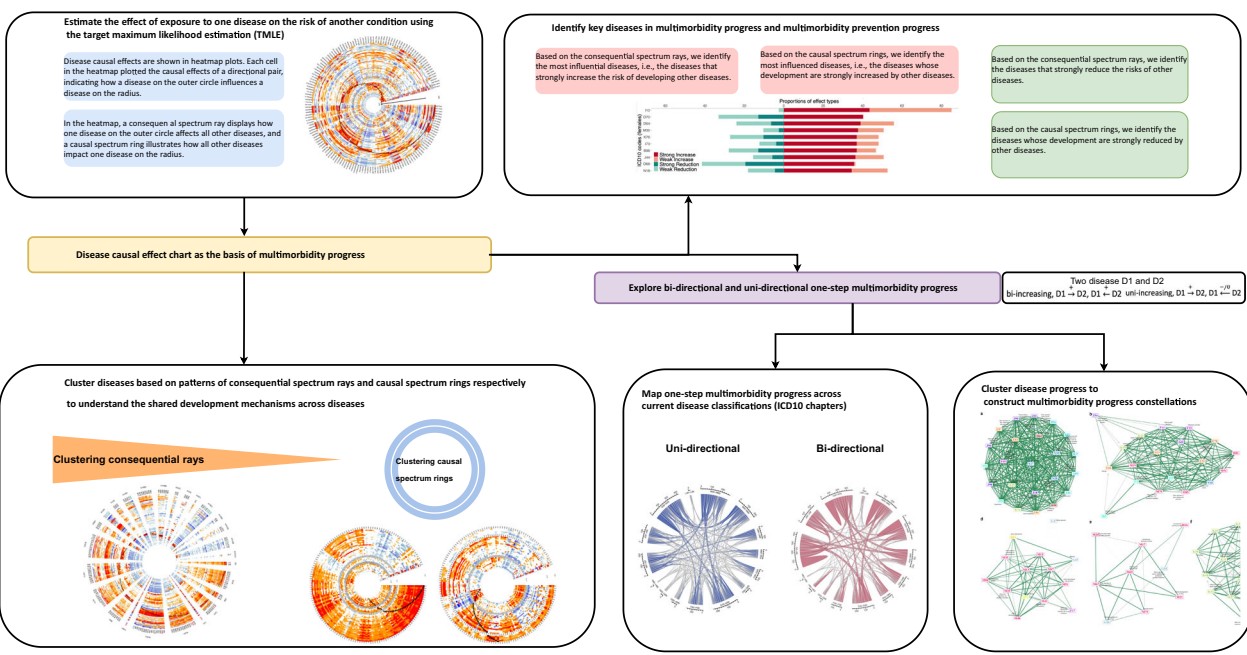

**Fig. 1 | Framework of mapping multimorbidity progression based on causal relationships between diseases.** A schematic of the study procedure with arrows pointing to the subsequent steps.

when either the outcome model or the propensity score model is correctly specified. The inference relies on the positivity assumption, which is thoroughly examined in the Supplementary Methods, and the unmeasured unconfoundedness assumption, the plausibility of which is discussed later. We conducted the estimation for females and males separately, resulting in a comprehensive assessment covering 23,562 directional pairwise causal effects of diseases ($154 \times 153$) for females and 25,440 directional pairwise causal effects of diseases ($160 \times 159$) for males. In order to assess the rapidity of disease impact, we carried out a sensitivity analysis using the same estimation method but limited the analysis to a one-year follow-up.

We ranked the key diseases based on how they affect the likelihood of others and how their developments are affected by others, and identified the top influential and influenced diseases. Full details of this analysis are in the Supplementary Methods. We used Ward's hierarchical clustering algorithm to group diseases based on the similarity of how they affect others, as well as the similarity of how others affect them. To better understand how multimorbidity progresses from one ICD-10 chapter to another ICD-10 chapter, we analyzed the likelihood of one-step multimorbidity paths across the current disease classifications, separately for bi-directional and unidirectional progress, and for females and males.

We used a self-tuning k-means clustering method to cluster one-step multimorbidity progress into multimorbidity progress constellations or clusters and used the Kamada-Kawai algorithm to create a visually appealing cluster visualization of disease nodes (Supplementary Methods). The algorithm assessed the quality and quantity of links to each node, providing a reliable estimate of their overall importance. We tested the stability of clustering by comparing our clusterings with clusterings from the traditional k-means method with different initial random seeds, and the self-tuning spectral clustering method. As an alternative sensitivity analysis, we tested the stability by comparing results across distinct clustering methods.

All the analyses were conducted separately for females and males to compare multimorbidity progress between sexes.

## Role of the funding source

The funders had no role in study design, data collection, data analysis, data interpretation, write of the report, or the decision to submit the paper for publication.

## Reporting summary

Further information on research design is available in the Nature Portfolio Reporting Summary linked to this article.

## Results

### Disease causal effect chart as the basis of multimorbidity progress

Disease causal effect heatmap plots are shown in Fig. 2. The heatmap pie featured a consequential spectrum ray, displaying how one disease on the outer circle affects all other diseases, and a causal spectrum ring, illustrating how all other diseases impact one disease on the radius. Each cell plotted the causal effects of a directional pair, indicating how a disease on the outer circle influences a disease on the radius. For females, 7135 (30.3%) directional pairs had increasing effects ($\xrightarrow{+}$, cells with red shades), 7330 pairs (31.1%) had decreasing effects ($\rightarrow$, cells with blue shades), and the remaining showed no significant effects (cells with white color). Out of all directional pairs evaluated for males, 7541 pairs (29.6%) had increasing effects, 7941 pairs (31.2%) had decreasing effects, and the remaining showed no significant effects. Patients who have been diagnosed with a disease starting the directional pairs with increasing effects have a higher chance of developing the other disease and experiencing multimorbidity or increasing their disease counts.

### Influential and influenced diseases

For each sex, we identified the top 10 influential diseases that are most likely to be associated with progression to other conditions (Fig. 3a). The top disease in females was hypertension (I10) while in males it was non-insulin dependent diabetes (E10), followed by hypertension. Five of these diseases were common to both sexes, including hypertension (I10), anaemias (D64),

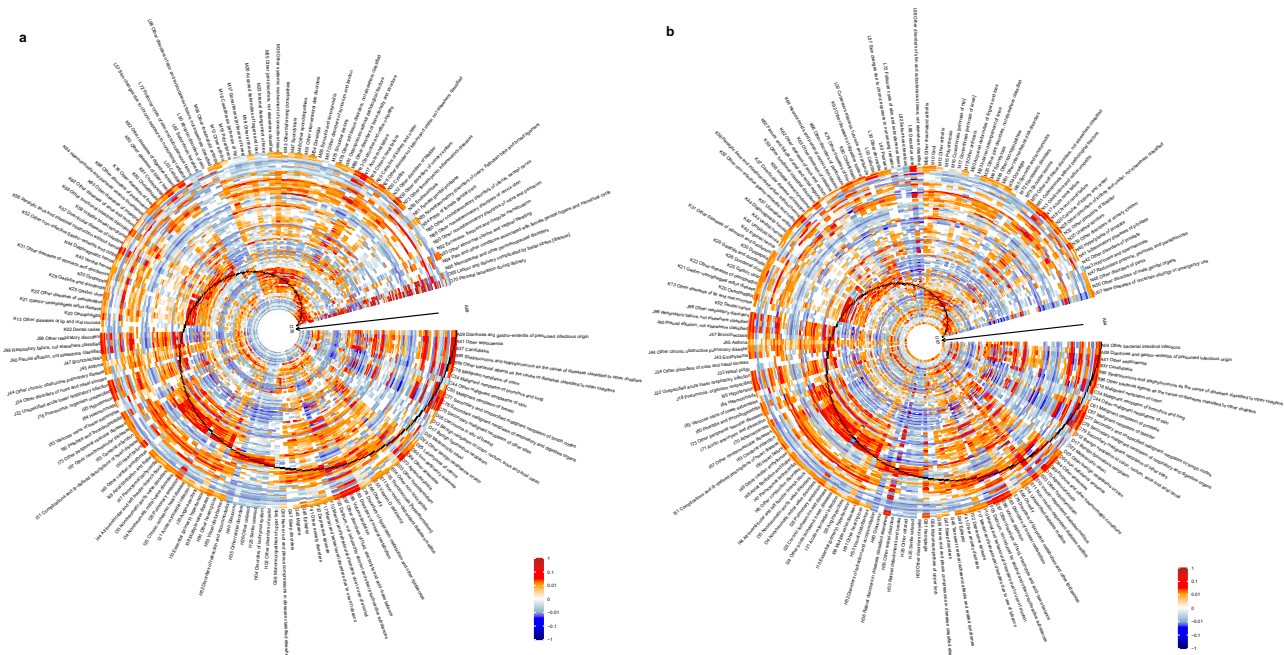

**Fig. 2 | Estimated pairwise causal effects among 190 diseases. a** females, estimates of 23,562 ($154 \times 153$) directional pairwise causal effects of diseases. **b** males, estimates of 25,440 ($160 \times 159$) directional pairwise causal effects of diseases. Each cell plotted the estimated causal effects of a directional pair, indicating how a disease on the outer circle influences a disease on the radius. 95% confidence intervals are shown in Supplementary Data 1 and 2. Cells in black represent diseases to themselves. The heatmap pie chart featured a consequential spectrum ray, displaying how a diagnosis on the outer circle affects all other diagnoses, and a causal spectrum ring, illustrating how all other diagnoses impact a diagnosis on the radius

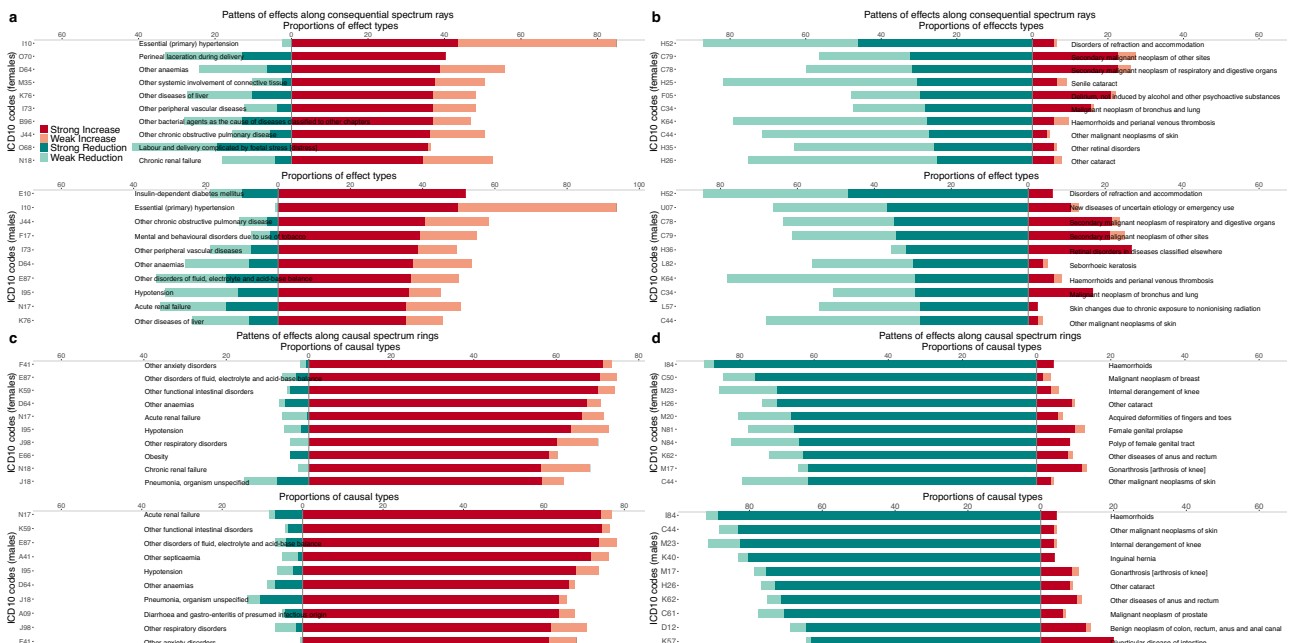

**Fig. 3 | Top diseases in multimorbidity progress and multimorbidity prevention progress.** Diseases are ordered by the proportions of other diseases affected by them (**a**, **b**). **a** Top 10 diseases in strongly increasing the likelihood of developing other diseases. **b** Top 10 diseases in strongly reducing the likelihood of developing other diseases. Diseases are ordered by the proportions of other diseases affecting them (**c**, **d**). **c** Top 10 diseases whose development is strongly increased by other diseases. **d** Top 10 diseases whose development is strongly reduced by other diseases.

liver diseases (K76), peripheral vascular diseases (I73), chronic obstructive pulmonary disease (COPD, J44); these diseases are most likely to cause both female and male patients with single diseases to progress to multimorbidity or with multimorbidity to increase disease counts. For females, the top 10 additionally included two female-specific diseases (labor and delivery complicated by foetal stress [O68] and perineal laceration during delivery [O70]), systemic involvement of connective tissue (M35), bacterial agents as the cause of diseases (B96), and chronic renal failure (N18). For males, the top 10 also included mental and behavioral disorders due to use of tobacco (F17), disorders of fluid, electrolyte and acid−base balance (E87), hypotension (I95), and acute renal failure (N17). Having these diseases is likely to greatly increase the occurrences of many other diseases, and socio-demographics, environments, and this may help explain why individual behavioral factors (smoking and alcohol use) that affect these diseases are likely to be associated with many other diseases[17–21].

Also, we identified the top 10 influenced diseases that are most likely to develop after initially having other diseases (Fig. 3c). Eight of these diseases were common to both sexes, including anxiety disorder (F41), disorders of fluid, electrolyte and acid−base balance (E87), functional intestinal disorders (K59), anemias (D64), acute renal failure (N17), hypotension (I95), respiratory disorders (J98), and pneumonia (J18). Additionally, we discovered that anemias (D64) strongly increased many other diseases, as well as being greatly increased by many other diseases in both sexes and that renal failure (acute renal failure [N17], chronic renal failure [N18]) and hypotension (I95) had the same characteristics in at least one sex.

During the study period, several of the top influential diseases (females, liver diseases [K76] and chronic renal failure [N18]; males, disorders of fluid, electrolyte and acid−base balance [E87], hypotension [I95], acute renal failure [N17], liver diseases [K76]) showed a quick effect, with over 50% of subsequent diseases occurring within a median time interval of less than one year after the onset of these influential diseases (Supplementary Fig. 6). In comparison, the top influenced diseases displayed a median time range of one to five years between their onset and the previous diseases, for over 60% of all previous diseases. Notably, two childbirth-related influential diseases, perineal laceration during delivery (O70) and labor and delivery complicated by foetal stress (O68), showed that more than 25% of subsequent diseases occurred more than five years after their onset.

Moreover, we identified the top 10 diseases that are less likely to be associated with progression to multimorbidity (Fig. 3b) and the top 10 diseases that are less likely to develop after initially having other diseases (Fig. 3d) for both males and females, which could be related to multimorbidity prevention. For the former, the top disease in both females and males was disorders of refraction and accommodation (H52). For the latter, the top disease in both females and males was haemorrhoids (I84), indicating that both female and male patients with a wide range of different diagnoses were less likely to develop haemorrhoids in current clinical practice, compared to those without the diagnoses (Supplementary Results).

### Clustering consequential spectrum rays and causal spectrum rings reveal shared multimorbidity development mechanisms

When using consequential spectrum rays to cluster diseases, we identified 26 clusters for females and 28 clusters for males (mean clustering significance, female $P_{au} = 0.87$, male $P_{au} = 0.88$). Diseases in the same cluster have the same pattern of affecting the risks of other diseases, likely reflecting shared consequences. Among females, the three largest clusters were the respiratory-dominant cluster (Cluster VII), the musculoskeletal-dominant cluster (VI), and the circulatory-dominant cluster (XII). For males, the top three clusters were the respiratory-dominant cluster (V), the circulatory-dominant cluster (IX), and the digestive-dominant cluster (X). These clusters contain diseases where having particular disease could increase the risks of a large group of diseases. Additionally, the largest cluster included diseases that could strongly increase the risks of developing many diseases and strongly decrease the risks of many diseases simultaneously (VII in a and V in b, Supplementary Fig. 1).

In a similar manner, we used causal spectrum rings to group diseases and identified 14 clusters for females and 14 clusters for males (mean clustering significance, female $P_{au} = 0.87$, male $P_{au} = 0.89$, Supplementary Fig. 2). Diseases in the same cluster have the same pattern of being affected by similar groups of diseases, likely reflecting shared etiology. We found a substantial difference between the largest clusters and the smallest clusters.

The largest cluster in both sexes was a large hybrid cluster covering multiple ICD-10 chapters and consisting of diseases that could be increased by most diseases involved in the study, although the diseases involved varied between sexes. For females, the largest cluster was mainly composed of diseases of malignant neoplasms, digestive, respiratory, metabolic, and genitourinary systems, and infectious and parasitic diseases (Cluster I). For males, it primarily included diseases of malignant neoplasms, digestive and respiratory systems, and infectious and parasitic diseases (Cluster I). The smallest cluster in both sexes consisted of diseases that were less likely to be increased by the risks of most diseases involved in the study.

Our spectrum ray clustering sheds light on how the top diseases contribute to the multimorbidity progress. We found that the top 10 influential diseases- those that strongly increase most other diseases as identified in the previous section, covered a broad spectrum of clusters in both sexes. The 10 diseases for females were classified into 9 clusters (Supplementary Fig. 1a) and the 10 diseases for males were classified into 6 clusters (Supplementary Fig. 1b). This means that the most influential diseases in multimorbidity progress are more likely to influence different groups of diseases, suggesting the dominant roles of influential disease in different multimorbidity mechanisms. On the other hand, the top influenced 10 conditions that were strongly increased by most other diseases, tended to center only in a few clusters and this finding was consistent in both sexes (Cluster I, Supplementary Fig. 2), indicating the shared etiology among the most affected diseases. Finally, our clustering adds to show how those diseases are unlikely to be associated with multimorbidity progress (Supplementary Results).

## Multimorbidity progress across current disease classifications

The study involved 16 ICD-10 chapters that group diseases into similar categories according to body systems (Supplementary Methods). Most of the uni-directional progress crossed the 16 ICD-10 chapters, with 964 out of 1080 (89.3%) for females and 1217 out of 1339 (90.9%) for males. However, such progress did not substantively differ from random crossings, except for the following cases in females: from the neoplasm chapter (C00-D48) to the infectious chapter (A00-B99) and the skin chapter (L00-L99), from the blood chapter (D50-D89) to the neoplasm chapter, from the mental chapter (F00-F99) to the respiratory chapter (J00-J99), from the nervous chapter (G00-G99) to the circulatory chapter (I00-I99), from the circulatory chapter to the infectious chapter, from the genitourinary chapter (N00-N99) to the digestive chapter (K00-K93), from the musculoskeletal chapter (M00-M99) to the circulatory chapter, and from the child-birth chapter (O00-O99) to the genitourinary chapter and the skin chapter, and several cases in males from the circulatory chapter to the newly emerging chapter (U00-U49) and the blood chapter, from the digestive chapter to the neoplasm and genitourinary chapters ($P_{perm}$ < 0.05, Supplementary Data 3, Fig. 4a, b).

When compared to uni-directional progress, bi-directional progress presents considerably clustering tendencies both within and across ICD-10 chapters. Notably, bi-directional progress is more likely to occur in the same ICD-10 chapters (Fig. 4c, d). This pattern is observable in different ICD-10 chapters, such as the circulatory, respiratory, digestive, musculoskeletal, and genitourinary chapters (I00-I99, J00-J99, K00-K93, M00-M99, N00-N99, $P_{perm}$ = 0.0001, Supplementary Data 3). Although a smaller proportion of strong bi-directional progress than uni-directional progress covers different ICD-10 chapters (1035 [66.6%] for females and 1134 [67.7%] for males), many now showed multimorbidity progress tendency in both sexes (neoplasm and skin chapters [C00-D48 and L00-L99], infectious and blood chapters [A00-B99 and D50-D89], blood and digestive chapters [D50-D89 and K90-K93], circulatory and respiratory chapters [I00-I99 and J00-J99], metabolic and mental chapters [E00-F90 and F00-F99], nervous and musculoskeletal chapters [G00-G99 and M00-M99], eye and ear chapters [H00-H59 and H60-H95], $P_{perm}$ < 0.05, Supplementary Data 3). In males, one-step progress was also observed between the circulatory chapter and the metabolic chapter and between the infectious chapter and the digestive chapter.

## Multimorbidity progress constellations

We identified 10 constellations for females (clustering stability, mean adjusted Rand index 0.76, Fig. 5) and 9 constellations for males (clustering stability, mean adjusted Rand index 0.75, Fig. 6). Higher adjusted Rand indexes indicate greater clustering stability. Adjusted Rand Indexes from sensitivity analyses were still sufficiently high, 0.65 and 0.68 for clusterings in females and males, respectively. Diseases in the same constellation are either directly causally linked or causally related through other diagnoses in the same constellation.

The first constellations with ferris wheel-like structures in females (Fig. 5a) and males (Fig. 6a) are characterized by strong bi-increasing connections between members, indicative of intricate disease trajectories across multiple ICD-10 chapters. Diseases in the two constellations largely overlap, including infectious diseases (diarrhoea and gastro-enteritis [A09], septicaemia [A41], candidiasis [B37], streptococcus and staphylococcus [B95], bacterial agents as the cause of diseases [B96]), anemias (iron deficiency anaemia [D50], anaemias [D64]), metabolic disorders (disorders of mineral metabolism [E83], volume depletion [E86], disorders of fluid, electrolyte and acid-base balance [E87]), delirium (F05), hypotension(I95), respiratory infections and diseases (pneumonia [J18], acute lower respiratory infection [J22], pleural effusion [J90], respiratory failure [J96], respiratory disorders [J98]), diseases of the genitourinary system (acute renal failure [N17], disorders of kidney and ureter [N28], disorders of urinary system [N39]), and diseases of digestive systems (functional intestinal disorders [K59], diseases of liver [K76]). These diseases are mostly bi-directionally connected, and thus the progressions of multimorbidity within the constellations are generally similar in both females and males. However, females may experience pulmonary embolism (I26), while males may also have uncertain new diseases (U07) in these trajectories of multimorbidity.

The hypertension-asthma constellations show a considerable number of uni-directional progress, but most of the diseases within them are still tightly clustered (Figs. 5b and 6b). Both females and males could experience cross-chapter trajectories that progress through hypertension (I10), asthma (J45), obesity (E66), disorders of lipoprotein metabolism and other lipidaemias (E78), gastro-oesophageal reflux disease (K21), anxiety and depression (depressive episode [F32], anxiety disorders [F41]), as well as various musculoskeletal and connective tissue disorders. Additionally, males may also experience hearing loss (H91) and dermatitis (L30) in these trajectories of multimorbidity, while females may experience peripheral vascular diseases (I73) and chronic renal diseases (N18).

The COPD-circulatory constellations have more uni-directional connections (Figs. 5c and 6c). The constellations revealed the multimorbidity trajectories among COPD and bronchiectasis (chronic obstructive pulmonary disease [J44], bronchiectasis [J47]), a variety of circulatory diseases, diabetes (E11), epilepsy (G40), cellulitis (L03), mental disorders due to alcohol (F10). However, mental disorders due to use of tobacco (F17) and hemiplegia (G81) are not involved in the female constellation, unlike in males.

In females, there were two constellations primarily consisting of the multimorbidity between neurological disorders and musculoskeletal and connective tissue disorders. The first was associated with the central nervous system (migraine [G43], sleep disorders [G47], Fig. 5d), while the second was linked to the peripheral nervous system (nerve root and plexus compressions [G55], mononeuropathies of upper limb [G56], Fig. 5e). In males, the former was combined with the COPD-circulatory constellation. Neurological disorders cluster with musculoskeletal and connective tissue diseases primarily through uni-directional progress, through bi-directional progress is possible (Fig. 6d).

The constellations related to the digestive system encompass a variety of bi-directional and uni-directional progress that can be classified into two distinct groups. In the first group, cross-chapter patterns commonly emerge among benign colon neoplasms (D12), haemorrhoids (I84), and several digestive diseases (Figs. 5f and 6e). In the second group, cross-chapter patterns may develop among malignant cancers (secondary and

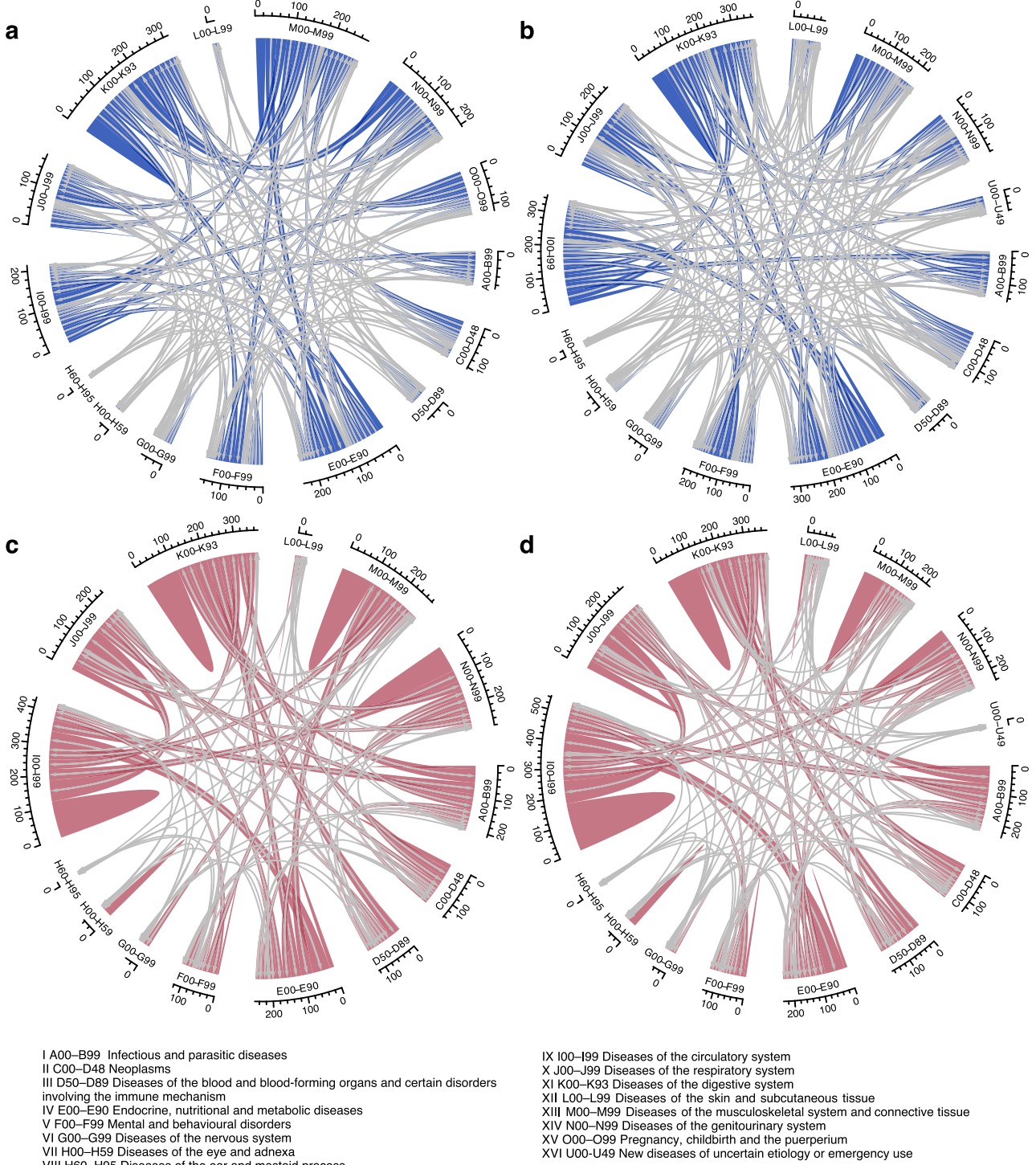

I A00–B99  Infectious and parasitic diseases
II C00–D48  Neoplasms
III D50–D89  Diseases of the blood and blood-forming organs and certain disorders involving the immune mechanism
IV E00–E90  Endocrine, nutritional and metabolic diseases
V F00–F99  Mental and behavioural disorders
VI G00–G99  Diseases of the nervous system
VII H00–H59  Diseases of the eye and adnexa
VIII H60–H95  Diseases of the ear and mastoid process

IX I00–I99  Diseases of the circulatory system
X J00–J99  Diseases of the respiratory system
XI K00–K93  Diseases of the digestive system
XII L00–L99  Diseases of the skin and subcutaneous tissue
XIII M00–M99  Diseases of the musculoskeletal system and connective tissue
XIV N00–N99  Diseases of the genitourinary system
XV O00–O99  Pregnancy, childbirth and the puerperium
XVI U00–U49  New diseases of uncertain etiology or emergency use

**Fig. 4 | Chord diagrams showing the patterns of one-step multimorbidity progress among ICD-10 chapters by sex and progress direction. a** females, uni-directional progress. **b** males, uni-directional progress. **c** females, bi-directional progress. **d** males, bi-directional progress. The blue arcs represent the amount of uni-directional one-step multimorbidity that progresses from one chapter to another chapter following the direction of the arrows. The red arcs represent the amount of bi-directional one-step multimorbidity that progresses between two chapters. The thicker the arcs, the larger the amount. Fragments going around the circumference represent the total amount of one-step multimorbidity in the chapter. Arcs without inner lines represent self-links. Only pairs with effect sizes larger than 0.01 are counted.

unspecified malignant neoplasm of lymph nodes [C77], secondary malignant neoplasm of respiratory and digestive organs [C78], secondary malignant neoplasm of other sites [C79]), agranulocytosis (D70), obstructive and reflux uropathy (N13), multiple peritoneum diseases (ventral hernia [K43], paralytic ileus and intestinal obstruction without

hernia [K56], disorders of peritoneum [K66]), and gallbladder disorders (K80, Figs. 5g and 6f).

There were also sex-specific constellations. For females, there are two constellations of genital constellations. One of them includes mainly uni-directional progressions from diseases related to childbirth (labor and

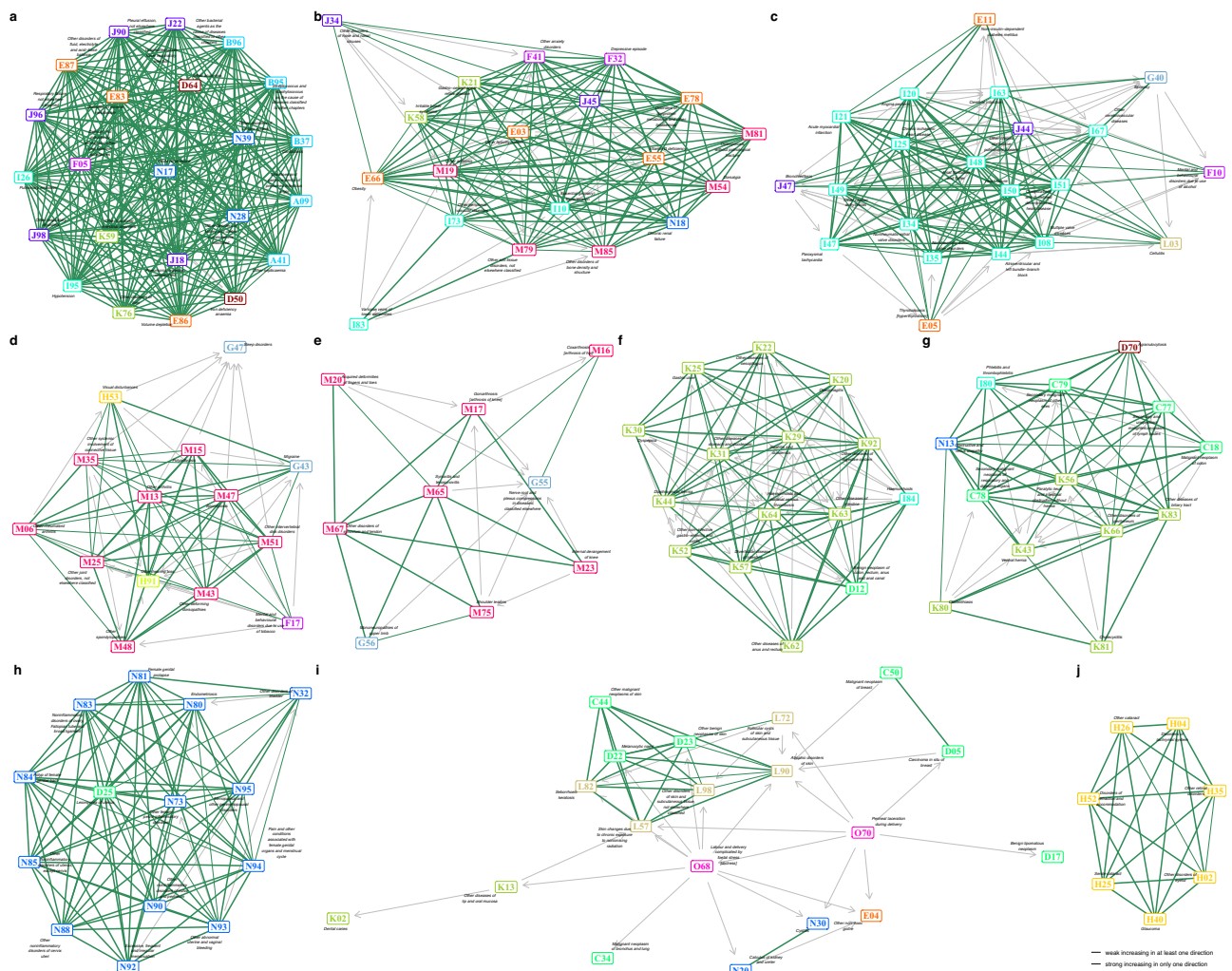

**Fig. 5 | Maps of the multimorbidity progress constellations among females. a–j.** different multimorbidity progress constellations. **a** strong bi-directionally connected constellation. **b** hypertension-asthma constellation. **c** COPD-circulatory constellation. **d** neurological-central-nervous constellation. **e** neurological-peripheral -nervous constellation. **f** digestive-benign-neoplasms constellation. **g** digestive-malignant-neoplasm constellation. **h** genital constellation. **i** genital-skin constellation. **j** eye constellation. Nodes represent the ICD-10 codes and different colors represent different ICD-10 chapters. Green segments represent bi-directional multimorbidity progress and gray segments represent uni-directional

multimorbidity progress with arrows pointing to the following diseases. The thicker the segments, the stronger the connections. Nodes that are more strongly connected are placed closer together. The 10 diseases, including disorders of fluid, electrolyte and acid-base balance (E87), hypertension (I10), chronic ischaemic heart disease (I25), deforming dorsopathies (M43), gonarthrosis [arthrosis of knee] (M17), diverticular disease of intestine (K57), secondary malignant neoplasm of respiratory and digestive organs (C78), noninflammatory disorders of cervix uteri (N88), malignant neoplasms of skin (C44), senile cataract (H25) are the hub diseases.

delivery complicated by foetal stress [O68] and perineal laceration during delivery [O70]) and breast cancer (malignant neoplasm of breast [C50], carcinoma in situ of breast [D05]) to skin diseases (skin changes due to chronic exposure to nonionising radiation [L57], follicular cysts of skin and subcutaneous tissue [L72], disorders of skin and subcutaneous tissue [L98]) and skin cancers (malignant neoplasms of skin [C44], melanocytic naevi [D22], benign neoplasms of skin [D23]), malignant neoplasm of bronchus and lung (C34), lipomatous neoplasm (D17), dental caries (K02), and diseases of the lip and oral mucosa (K13, Fig. 5i). Interestingly, males have a similar constellation, but instead of childbirth-related diseases, it includes type I and unspecific diabetes (insulin-dependent diabetes mellitus [E10], diabetes mellitus [E14], Fig. 6h). The other female genital multimorbidity constellation includes bi-directional progressions between leiomyoma of the uterus (D25) and many female genitourinary diseases, while the other male genital multimorbidity constellations include many male genitourinary diseases, as well as malignant neoplasm of the prostate and bladder (malignant neoplasm of prostate [C61], malignant neoplasm of bladder [C67], Figs. 5h and 6g).

Finally, we identified the 10 hub diseases for 10 female constellations, and the 9 hub diseases for 9 male constellations, and four female constellations have the same hub diseases as male constellations: hypertension-asthma (hypertension [I10]), COPD-circulatory (chronic ischaemic heart disease [I25]), genital (malignant neoplasms of skin [C44]), and eye (senile cataract [H25]).

## Discussions
In this study, we examine how multimorbidity progresses among 190 diseases by examining the comprehensive causal links among them and analyzing their coalescing patterns. Whereas previous studies that investigated multimorbidity trajectory and clusters focused on disease associations, we clustered relying on the causal relationships among them, which, to our knowledge, is the first in-depth description of the causal progression of multimorbidity for hundreds of diseases. We also compared the progression in males and females and found important differences. Importantly, our approach is capable of accounting for bi-directional causal relationships among diseases, which could enable the discovery of complex trajectory

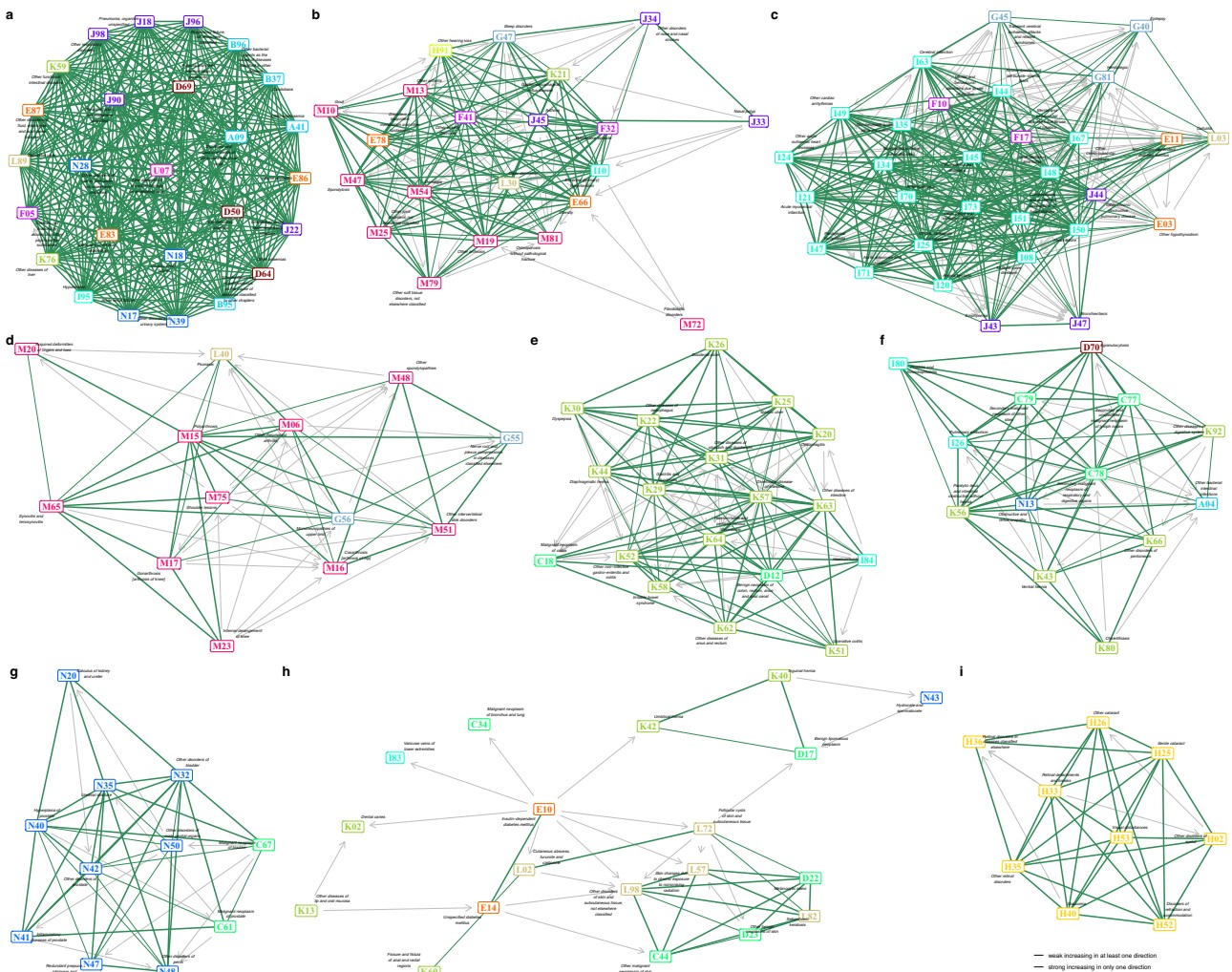

**Fig. 6 | Maps of the multimorbidity progress constellations among males. a–i** different multimorbidity progress constellations. **a** strong bi-directionally connected constellation. **b** hypertension-asthma constellation. **c** COPD-circulatory constellation. **d** neurological-peripheral-nervous constellation. **e** digestive-benign-neoplasms constellation. **f** digestive-malignant-neoplasm constellation. **g** genital constellation. **h** diabetes-skin constellation. **i** eye constellation. Nodes represent the ICD-10 codes and different colors represent different ICD-10 chapters. Green segments represent bi-directional multimorbidity progress and gray segments represent uni-directional multimorbidity progress with arrows pointing to the following diseases. The thicker the segments, the stronger the connections. Nodes that are more strongly connected are placed closer together. The 9 diseases, including acute renal failure (N17), hypertension (I10), chronic ischaemic heart disease (I25), intervertebral disk disorders (M51), diverticular disease of intestine (K57), secondary and unspecified malignant neoplasm of lymph nodes (C77), hyperplasia of prostate (N40), malignant neoplasms of skin (C44), senile cataract (H25) are the hub diseases.

networks among diseases and identify common, mechanistically related, and prognostically relevant clusters. Our developed framework and tools are an important step towards meeting the challenge of mapping the progression of multimorbidity.

We provide a clinically driven comprehensive list of key diseases to be included when addressing multimorbidity and show that lists between females and males largely overlap. The profiles help healthcare providers in recognizing the risk of subsequent diseases while assessing patients without multimorbidity and in determining intervention and prevention strategies to prevent multiple long term conditions[10]. We emphasized this profile should be cautiously interpreted with data limitations we shall discuss later. Whether the identified causal relationships between two diseases are a result of the natural development mechanism of multi-morbidity, clinical treatment practice for the initial diagnosis, patients' adapting to healthier lifestyles after the initial diagnosis, or if screening strategies require further investigation. The top influential and influenced disease lists included acute diseases, emphasizing their significance in preventing and managing chronic multimorbidity. Many top influential diseases showed a quick effect. A secondary analysis revealed that a

majority of the diseases (86.9% for females and 89.1% for males) displayed a heightened causal effect within the first year of follow-up (Supplementary Fig. 8). This could be attributed to the fact that illnesses that develop after the initial diagnosis are more prone to be detected and diagnosed within a shorter timeframe, rather than over a more prolonged period (Supplementary Data 4 and 5). It remains unclear whether this reflects the natural development mechanism of multimorbidity or if screening strategies warrant further investigation.

By clustering consequential spectrum rays and causal spectrum rings, we can gain insights into shared consequences and etiology among diseases. This approach allows for a systematic view to update disease classification according to their connections and discover potential shared treatments. For instance, the clustering of rays confirms the well-established link between asthma and hypertension[22], and supports the association of these conditions with arthritis[23]. This similarity in their effects on other diseases suggests that potential shared therapy strategies, including re-purposed approved drugs for one disease, could be useful in treating these diseases. It is worth noting that the most influential diseases contributing to multi-morbidity span a wide variety of identified ray clusters in both men and

women, suggesting complex mechanisms at play. Meanwhile, the most affected diseases tend to center around only one ring cluster, indicating that shared underlying causes may exist among the most affected conditions.

Currently, healthcare concentrates on treating specific diseases or organ systems, which can result in overlooking opportunities to address multiple long term conditions simultaneously[24]. We have identified both bi-directional and uni-directional one-step progression of multimorbidity. The former displays substantial clustering tendencies within and across International Classification Disease chapters, dominating most identified patterns of multimorbidity progress constellations. This groundwork can inform future studies aimed at developing customized strategies for preventing and treating multimorbidity. Identifying patterns of cross-chapter multimorbidity progress is an essential initial step toward targeted integrated and person-centered care for individuals with unmet needs[7]. The identified patterns provide evidence for reevaluating funding, research, training, and treatment across different specialties to address this growing concern of multimorbidity[4,25].

Multimorbidity progress constellations reveal intricate patterns that can shed light on the immune system's involvement in this complex condition. Inflammation mechanisms and shared molecular pathways appear to be related to these patterns[26]. In both sexes, the most strongly connected multimorbidity progress constellations involve diseases caused by inflammation from infection[27,28], including anemia due to a conserved defense strategy of the body directed against invading microbes[29], which are bi-directionally linked with liver diseases, renal diseases, and metabolic disorders[8]. The hypertension-obesity-asthma constellations involve diseases that stem from dysregulation of the inflammatory response, not caused by infection[30]. The COPD-cardiovascular constellations appear to be a mixture of bacterial-induced inflammation, such as cellulitis and airway inflammation-driven COPD, and low-grade systemic inflammation in COPD, diabetes, and cardiovascular diseases[31–34]. These two constellations include a number of mental health problems that could be attributed to the pro-inflammatory cytokine theory of mental conditions where the immune system subjugates the brain[35]. Digestive-related constellations are further divided into two subclusters. One subcluster is linked to inflammation caused by infections, including benign cancers or inflammatory pseudotumor[36]. The other subcluster is associated with immune mechanisms in malignant cancer and agranulocytosis where the role of the immune system is increasingly evident[37,39]. Furthermore, eye diseases seldom cluster with other diseases, with visual disturbances exceptions. While these findings point toward inflammation mechanisms as a plausible determinator of patterns of multimorbidity progress, they are hypothesis-generating only and require further assessment. Additionally, because inflammation is sensitive to socioeconomic, environmental, and behavioral risk factors[33–35,40], confirming this hypothesis could help to explain inter and intra-population differences in multimorbidity across life stages, ethnicities, sexes, socioeconomic groups, and geographies[41–44].

Previous studies have suggested that neurological disorders could cause pain in certain areas or affect joints, muscles, and bones[45]. On the other hand, musculoskeletal disorders and connective tissue diseases can present with neurological symptoms[46,47]. Our findings indicate a potential bi-directional progress between these two categories of conditions, affirming established perspectives and implying a profound interconnection between these illnesses.

The observed patterns are generally similar between females and males, but there are also important differences. Note that we used the same methods for females and males and have controlled socio-demographic factors and healthcare utilization, the findings on sex-specific differences are more likely to represent the biological differences. The differences in sex-specific constellations such as bladder cancer and breast cancer, are apparent[48,49]. Also, tobacco-induced mental conditions are closely connected with COPD and diabetes in males, but not in females, possibly due to sex differences in inflammation

and immune dysregulation[50]. We have discovered that sleep disorders and hearing loss, both of which have been linked to Alzheimer's disease[51,52], cluster within hypertension-obesity-asthma constellations in males, while musculoskeletal disorders and connective tissue diseases in females. This points to different mechanisms and pathways in Alzheimer's disease in females and males[53,54].

Mechanisms underlying the development of multimorbidity are complex and multilevel, and genetic predispositions and mechanisms could also be at play[14,55]. Previous studies have shown that multiple long term conditions that affected the same physiological system showed a greater propensity for sharing loci-level genetic components, whereas those affecting different systems had a greater probability of sharing network-level genetic components[14]. Diseases in the same sex-specific constellations tend to share genes[14,56]. Further studies demonstrate that the influence of genetics on the development of diseases may vary between males and females[57]. By pinpointing distinct disease patterns in both sexes, we can delve into the shared biological origins of sex-specific constellations and genetic mechanisms in the development of multimorbidity.

The study is an observational study, which can be subject to potential confounding bias[58,59]. Although we have taken measures to control for known confounders and have identified statistically significant effects between many pairs of diseases, there may still be alternative explanations. It is possible that there are unmeasured factors that could also be responsible for the observed causal links, and these have not been accounted for in our data. This includes diseases that occurred in early childhood, unmeasured environmental exposures, common genetically determined mechanisms, and current treatment practices. A data-rich and thoroughly phenotyped dataset with treatment details is required to conduct address these issues. Nevertheless, our findings generate hypotheses that may be tested in a smaller, well-characterized sample.

It's important to note that our dataset is limited to a single country, and as such, the findings may not be applicable to other countries that have substantial differences in unmeasured confounders compared to the United Kingdom. We also excluded rare diseases, which could limit our understanding of how these diseases may be connected with more common diseases. Additional research is required to determine the extent to which our findings can be generalized to other countries and the diseases that were excluded.

Also, it's possible that the study contains inaccuracies in measurements, such as certain conditions that are presently undetectable or undiagnosed in clinical settings, which could lead to biased estimates. To address this challenge, it's crucial to gather reliable data from population-based cohort studies that conduct comprehensive and regular assessments of individual disease status throughout the duration of the study.

As a statistical classification, the ICD-10 code has linearization properties, notably mutual exclusivity of categories and exhaustive coverage of the domain of interest. However, disease conditions are often complex and can involve multiple parts of the body. As a result, some conditions in the ICD-10 code may not be located in appropriate chapters, which can lead to incorrect counting of multimorbidity progress within and across chapter-level categories. In fact, our identified patterns of cross-chapter multimorbidity progress could help investigate such potentially inappropriate classification, where higher clustering cross-chapter tendency may indicate relationships of certain diseases to both chapters. Furthermore, the ICD-10 code may not accurately capture emerging diseases, which are classified under the new diseases of uncertain etiology or emergency use chapter (U00-U49). However, these emerging diseases are typically less prevalent and thus are not the focus of the study.

By examining comprehensive causal links among diseases and analyzing their coalescing patterns, our results characterize multimorbidity progress among the most prevalent diseases. The map of multimorbidity progress and its clustering between sexes provides important information

on understanding multimorbidity development mechanisms, as well as designing targeted community-based multimorbidity screening for the ongoing Life Cohort Study in China[60] and reconfiguring services to better meet patients' needs.

## Data availability

UK Biobank data can be requested directly from the UK Biobank project site, after successful completion of registration and the application process. Full details can be found: https://www.ukbiobank.ac.uk/. Source data are provided at https://github.com/ShashaHan-collab/MapMultimorbidityCluster[61].

## Code availability

Code for analyzing and visualizing the data is available at https://github.com/ShashaHan-collab/MapMultimorbidityCluster[61].

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

## Acknowledgements
The study was supported by the National Natural Science Foundation of China (No. 82304269), the Chinese Academy of Medical Sciences (CAMS) Innovation Fund for Medical Sciences (No. 2023-I2M-3-008), the Non-profit Central Research Institute Fund of Chinese Academy of Medical Sciences (No. 2023-ZHCH330-001), and the National Council for Scientific and Technological Development – CNPq (process number 308772/2022-9). The funders had no role in study design, data collection and analysis, decision to publish or preparation of the manuscript.

## Author contributions
S.H., R.S., and C.W. conceived of the research idea. All authors contributed to the study design. S.H., S.L., and Y.Y. carried out data analysis, model development, model validation, and data visualization. L.L., L.M., F.S.M., C.R.B., and C.W. provided expertise on disease pathology. L.L., L.M., Z.L., F.S.M., C.R.B., B.P.N., J.J.M., W.Y., R.S., and W.C. provided expertise on epidemiology and health science and assisted with the interpretation of results. S.H. prepared the first draft of the paper. All authors critically reviewed the manuscript and approved the final submission.

## Competing interests
The authors declare no competing interests.
