## [Peer Review File · Communications Medicine]

This manuscript has been previously reviewed at another Nature Portfolio journal. This document only contains reviewer comments and rebuttal letters for versions considered at *Communications Medicine*.Reviewers' comments:

Reviewer #1 (Remarks to the Author):

Referee Report Regarding the Manuscript:

COMMSMED-24-0060-T

Mapping multimorbidity progression among 190 diseases

Submitted to:

Communications Medicine

A. Brief Summary

This article presents a comprehensive study on the progression of multimorbidity among 190 diseases using data from over 500,000 UK Biobank participants over 12.7 years. Utilizing a machine learning method for causal inference, the study identifies key diseases that are influential in the progression to other conditions and those that are most likely to develop after initially having other diseases. The research team conducted analyses to explore bi-directional and uni-directional progress between diseases, utilized clustering analysis and visualization algorithms to identify multimorbidity progress constellations, and compared progression between sexes, unveiling distinct patterns and constellations for males and females.

B. Overall Impression and Recommendation

Despite its valuable contributions, the study suffers from several critical issues that require significant revisions and clarifications, as detailed later. In summary, while the study addresses a critical and complex issue in public health, it is imperative that the authors undertake significant revisions and provide further clarifications to strengthen the manuscript. Addressing the identified critical issues will not only enhance the reliability and robustness of the findings but also ensure that the study makes a meaningful contribution to our understanding of multimorbidity progression. Therefore, I recommend "major revision."

C. Major Comments

1. Assumptions of the TMLE Method: While TMLE is a powerful tool for causal inference, its application requires careful consideration of underlying assumptions, such as the correct specification of the outcome and treatment models, the assumptions of no unmeasured confounders, and the positivity assumption. The manuscript does not sufficiently address how these assumptions were verified or the potential impact of any violations on the study findings. A critical discussion on the sensitivity of the results to these assumptions, including any diagnostic tests performed or sensitivity analyses conducted, would significantly strengthen the conclusions' validity.
2. Selection of Diseases: The criterion for selecting diseases based on a prevalence rate greater than 1% in each sex could bias the analysis towards more common diseases and overlook the impact of rarer diseases that might have significant roles in multimorbidity progression. This selection criterion also risks oversimplifying the complex nature of multimorbidity, where less prevalent diseases could serve as pivotal connectors in the network of disease progression. The justification for this threshold is insufficient, and the article lacks a discussion on the potential impact of excluding rarer diseases on the study's findings.
3. Impact of Treatment and Healthcare Access: The study's analysis of disease progression patterns does not appear to account for the variability in treatment efficacy or access to healthcare services, which can significantly influence the course of multimorbidity. Differences in healthcare access and treatment responses could confound the relationships between diseases. Yet, there is no discussion on how these factors were considered in the analysis or how they might affect the interpretation of the results. Including such considerations is crucial for understanding the real-world applicability of the findings.
4. Methodological Clarity and Reproducibility: The manuscript overviews the analytical methods but lacks detailed explanations of the machine learning algorithms used, the rationale for their selection, and the parameter settings. For such a complex analysis involving machine learning and TMLE, it is essential to provide sufficient methodological detail to ensure the reproducibility of the findings. Furthermore, the article mentions using an appendix for supplementary analyses but does not integrate these supplementary results effectively into the discussion of the main findings.
5. Generalizability of Findings: The study is based on UK Biobank data, which, while extensive,

may not fully represent the global diversity in disease progression patterns due to geographic, ethnic, and socioeconomic variations. The manuscript briefly mentions this in the limitations but does not critically evaluate the potential biases introduced by relying on a single-country dataset. A more thorough discussion of how these limitations might affect the applicability of the findings to other populations and settings would make the limitations section more robust.

6. Validation of Clustering Methods: While the manuscript details the use of self-tuning k-means clustering and Kamada-Kawai visualization for identifying multimorbidity progress constellations, there is insufficient discussion on validating these clustering results. Given the potential for clustering algorithms to yield varying results depending on parameter settings and initial conditions, a rigorous validation approach is necessary, possibly including stability analysis or comparison with other clustering methods. Such validation is crucial for ensuring the reproducibility and generalizability of the identified multimorbidity patterns.

7. Handling of Sex-Specific Differences: The study highlights interesting sex-specific differences in multimorbidity patterns but does not sufficiently explore the biological or sociodemographic factors that might explain these differences. A more critical analysis of whether these differences could be attributed to methodological biases, differential healthcare utilization, or genuine biological distinctions would provide deeper insights into the observed patterns.

8. Assessment of Disease Classification System: The manuscript utilizes the International Classification of Diseases (ICD-10) for categorizing diseases, which is a standard approach. However, the study does not critically assess the limitations of this classification system in capturing the nuances of multimorbidity progression, especially for diseases that might straddle multiple categories or for emerging diseases not well-represented in ICD-10. A critical discussion on the implications of these classification challenges for the study's findings would enhance the manuscript's depth.

D. Minor Comments

1. Typographical Errors: The manuscript contains minor typographical errors (e.g., inconsistencies in disease names and slight grammatical errors) that should be corrected for readability.

2. Figure and Table Clarity: While the figures and tables are generally informative, some could benefit from higher resolution or clearer labeling to enhance their interpretability.

Reviewer #2 (Remarks to the Author):

Thank you for the opportunity to review the manuscript by Dr. Han et al titled "Mapping multimorbidity progression among 190 diseases"

In this work, the authors used the rich UK BioBank data set (500,000 participants 37-73 years, mean 57 and mean follow-up of 12.7 years)

The study aims to describe the progression of multimorbidity and map clusters of this progression; this entailed a complex matrix of correlations and other analysis in a pairwise fashion also to identify "influential" and "influenced: diseases as central point driving multimorbidity and multimorbidity pathways. One of the biggest claims of this study compared to previous publications is that Dr. Han et al provide their models' causal relationship rather than just association relationship.

The idea and concept are quite attractive: discover the natural progression of diseases in the context of multimorbidity, identify the influencer, direct attention to it, and prevent progression to more complex clusters of irreversible diseases. This is the holy grail of multimorbidity prevention and public health. The number of participants is large.

Major comments and overall impression:

1. The analysis and report are extremely complex with pretty illustrations but missing practical (clinical) interpretation. This is the most salient drawback, the project is ambitious and mathematically correct, however how as a clinician, health care policy maker, researcher, etc translate this in practical terms? This comment relates to several observations:

a. It is quite ambitious to map all disease trajectories, therefore you ended up with ICD-10 codes that have little meaning to the reader. ICD-10 codes sometimes have funny interpretations and ambiguous meanings to the clinician.

b. The graphs are quite beautiful but with little meaning.

2. If disease trajectories are the central point of your study, the data set you are using and the follow-up time and time to the association are not the appropriate ones:

- a. The average age is 57 years in your cohort: in the UK > 60% of your cohort have at least one comorbidity. Therefore, if you want to establish trajectories for chronic diseases you must study a pediatric population followed through the life course, which is not the case here.
- b. Many chronic diseases initiate at an early age, have a long latency period and manifest at mid and old age.
- c. Diseases with higher prevalence and easier to measure (blood pressure is easier than having a chest CT to detect lung cancer) will skew any multimorbidity analysis; in the your methods a thorough evaluation is performed on participants at enrollment, but encounters with the health systems obtain the follow-ups: the latter creates the above bias= at every encounter we will measure the blood pressure, therefore increasing the chance to diagnose hypertension, but less likely to detect diseases that need sophisticated studies: echo, chest CT, dexta-scans, blood markers etc. Therefore, the trajectory described reflects how UK physicians practice and order tests rather than a natural progression of diseases. To address this issue, you must measure a comprehensive battery of tests and questionnaires at established interval times to detect and establish the true prevalence.
- d. to establish a disease leading to another disease a period of one year is not adequate.

To illustrate my comments, on page 11 lines 204-211 of the main manuscript, the authors describe two entities that "prevent" disease progression. as the authors wrote, can help with "multimorbidity prevention": should I recommend my patients to wear glasses and have haemorrhoids? This paragraph summarizes the mathematical-to-clinical disconnect that I described above

Minor issues:

Please provide a table 1

In your graphs substitute ICD-10 codes and use common diseases names for ease of interpretation.

Responses to Reviewers' specific comments

Reviewer #1 (Remarks to the Author):

Referee Report Regarding the Manuscript:

COMMSMED-24-0060-T

Mapping multimorbidity progression among 190 diseases

Submitted to:

Communications Medicine

A. Brief Summary

This article presents a comprehensive study on the progression of multimorbidity among 190 diseases using data from over 500,000 UK Biobank participants over 12.7 years. Utilizing a machine learning method for causal inference, the study identifies key diseases that are influential in the progression to other conditions and those that are most likely to develop after initially having other diseases. The research team conducted analyses to explore bi-directional and uni-directional progress between diseases, utilized clustering analysis and visualization algorithms to identify multimorbidity progress constellations, and compared progression between sexes, unveiling distinct patterns and constellations for males and females.

Response: Thank you for this concise summary of our work.

B. Overall Impression and Recommendation

Despite its valuable contributions, the study suffers from several critical issues that require significant revisions and clarifications, as detailed later. In summary, while the study addresses a critical and complex issue in public health, it is imperative that the authors undertake significant revisions and provide further clarifications to strengthen the manuscript. Addressing the identified critical issues will not only enhance the reliability and robustness of the findings but also ensure that the study makes a meaningful contribution to our understanding of multimorbidity progression. Therefore, I recommend "major revision."

Response: We sincerely appreciate your recognition of the "valuable contributions" our work has made towards addressing "critical and complex issues in public health". Your feedback is highly valued, and we have thoroughly reviewed your suggestions to ensure that we have addressed your concerns in this revised version. We hope that the revised version meets your expectations. Thank you once again for taking the time to share your feedback with us.

C. Major Comments

1. Assumptions of the TMLE Method: While TMLE is a powerful tool for causal inference, its application requires careful consideration of underlying assumptions, such as the correct specification of the outcome and treatment models, the assumptions of no unmeasured confounders, and the positivity assumption. The manuscript does not sufficiently address how these assumptions were verified or the potential impact of any violations on the study findings. A critical discussion on the sensitivity of the results to these assumptions, including any diagnostic tests performed or sensitivity analyses conducted, would significantly strengthen the conclusions' validity.

Response: We would like to thank the reviewer for her/his useful input. We fully agree with the reviewer that these underlying assumptions are important for identification and estimation in causal inference.

We apologize for the lack of clarity in the original manuscript, which has now been amended (Methods Lines 135-139 “*TMLE is a doubly robust estimator, allowing for unbiased inference when either the outcome model or the propensity score model is correctly specified. The inference relies on the positivity assumption, which is thoroughly examined in the appendix (pp 6–8) and unmeasured unconfoundedness assumption, the plausibility of which is discussed later.*”).

The positivity assumption, or the experimental treatment assignment assumption, requires that propensity scores are in the range of (0, 1) for all possible treatment assignments almost everywhere. Intuitively, this assumption guarantees enough samples in both treated and untreated (or exposed and unexposed) groups at all levels of baseline covariates, ensuring the information for the corresponding potential outcome is accessible. We provided the number of samples for exposed and unexposed groups (Supplementary Table 5) to show there are sufficient samples for each directional pair of diseases. However, even if the assumption is valid for the true data-generating distribution, the randomness in data generating/sampling might cause a practical violation, jeopardizing the causal estimator's finite sample performance. One consequence of such violations is extreme values in the estimated propensity score. To address this issue, we followed a common approach and trimmed individuals with propensity scores outside a common range, as in the literature (Stürmer et al. 2001 [Ref 4, Supplementary Materials]), and clarified this in the appendix.

Appendix Lines 184-191: “*Violations of positivity assumptions could lead to extreme values in the estimated propensity score. We followed the common approach and trimmed individuals with propensity scores outside a common range, which is formed by the lowest propensity scores in the treated (having the exposure disease) individuals and the highest propensity scores in the control individuals (not having the exposure disease).⁴ Propensity scores were reestimated after the trimming, and the TMLE method was redone. In cases where propensity scores fell below 0.01 or above 0.99, we performed a second trimming using the cutoffs of the 5th percentile in the treated individuals and the 95th percentile in the control individuals.⁴”*

For the unmeasured cofoundedness assumption, however, we would like to clarify that there is no diagnostic test to test for it. The reason is that “data are not directly informative about the distribution of control outcome [status of disease D_{outcome} without having disease D_{exposure}] for those who received the active treatment [having disease D_{exposure}], nor are they informative about the distribution of the active treatment outcome [status of disease D_{outcome} after having disease D_{exposure}] given the receipt of the control treatment [without having disease D_{exposure}]. ” (Guido and Rubin 2015, p479 [Ref 58]). However, increasing the richness of the set of baseline covariates, their number and type, is likely to increase the credibility of the assumption (Rosenbaum 2002, Rubin 2009). As the other Reviewer critically pointed out, UK Biobank is a rich dataset; with this rich dataset, we have included substantially informative covariates in the datasets; in particular, we have included sociodemographic factors, health behaviors, health status, doctor-diagnosed disease, family history and all disease status and factors influencing health status and contact with health services at baseline status as confounders. That being said, in this revision, we have also strengthened the discussion of the potential impact of unmeasured confounders on the findings as follows:

Lines 474-485: “*The study is an observational study, which can be subject to potential confounding bias.^{58,59} While we have made every effort to control for measured confounders and found significant effects between many pairs of diseases, alternative explanations are possible when the unmeasured unconfoundedness assumption is not satisfied. There could be unmeasured shared etiology that may also be responsible for the observed causal links that are*

not accounted for in our data, such as diseases that occurred in early childhood, unmeasured environmental exposures, and common genetically determined mechanisms. Furthermore, our data does not account for the contributions of biological mechanisms and current treatment practices to the causal links among diseases. To conduct such causal and mechanistic investigations, a data-rich and thoroughly phenotyped dataset with treatment details is required. Nevertheless, our findings generate hypotheses that may be tested in a smaller, well-characterized sample.”

2. Selection of Diseases: The criterion for selecting diseases based on a prevalence rate greater than 1% in each sex could bias the analysis towards more common diseases and overlook the impact of rarer diseases that might have significant roles in multimorbidity progression. This selection criterion also risks oversimplifying the complex nature of multimorbidity, where less prevalent diseases could serve as pivotal connectors in the network of disease progression. The justification for this threshold is insufficient, and the article lacks a discussion on the potential impact of excluding rarer diseases on the study's findings.

Response: The Reviewer is correct that missing these rare diseases could prevent the study of how these diseases are clustered with more common diseases. The threshold is roughly determined based on similar approaches for disease association studies with large population-based cohort studies (Jensen et al. 2014; Westergaard et al. 2019). In Jensen et al. 2014, diseases with P-values for the relative risks greater than 0.001 were included, but they faced computational issues that made it impossible to perform the full procedure for estimating P-values (they instead used approximated values). Similarly, we face computational problems if we attempt to analyze the full 1,938 diseases. In Westergaard et al. 2019, diagnoses at the ICD-10 level 3 that occurred in more than 100 patients were excluded to avoid the same problem.

Unlike the two studies on disease associations, our objective is to investigate causation, and we excluded patients with a history of the two diseases before baseline or those with the outcome disease occurring before the exposure disease from the estimation (Appendix lines 158-160). As such, our current study design for causal inference may not be appropriate for studying the comorbidity of rare diseases. A more focused, smaller-sample study, such as a nested-control design with rare diseases considered as exposure, may be more appropriate for studying rare diseases.

Regarding the reviewer's concern that rare diseases might serve as pivotal connectors in the network of disease progression, this is less likely to be the case in our analysis. Note that for a rare disease to serve as an important connector, it must be that both a) several more prevalent diseases strongly increase its development (effect size ≥ 0.01 , Appendix lines 198-205, lines 255-260, lines 274-280) and b) its development strongly increases the likelihood of several more prevalent diseases (effect size ≥ 0.01). In fact, a) is likely to be true for a rare disease. To illustrate this, we have selected the first 100 less prevalent diseases and estimated the effect of having more prevalent diseases (the top 10) on the likelihood of developing them. We did not find any main effects larger than 0.01 when the current estimation methods were applicable. Additionally, the current methods might not be applicable due to severe violation of positivity assumptions for TMLE. For simplicity, we illustrate this for males, and the conclusions hold the same for females. The obtained results are shown in the Table below:

Table: The causal effect of having the prevalent disease hypertension (I10) on developing the first 100 less prevalent diseases than 1% for males. Outcome disease names were provided at the end of the Response document. The results for other prevalent diseases can be found in the linked file.

Exposure	Outcome	Main effects	Lower 95% CIs	Upper 95% CIs	P values
I10	G43	0.007	0.005	0.008	0.00E+00
I10	M67	0.001	0.000	0.002	7.04E-03
I10	M35	0.004	0.003	0.004	0.00E+00
I10	M85	0.003	0.002	0.004	0.00E+00
I10	E55	0.007	0.006	0.008	0.00E+00
I10	H04	0.002	0.002	0.003	0.00E+00
I10	R15	0.003	0.003	0.004	0.00E+00
I10	L90	0.001	0.000	0.002	1.40E-02
I10	M43	0.002	0.002	0.003	0.00E+00
I10	E05	0.002	0.002	0.003	0.00E+00
I10	E04	0.001	0.001	0.002	8.00E-07
I10	G20	0.002	0.001	0.002	2.53E-03
I10	K35	0.000	0.000	0.001	2.31E-01
I10	J30	0.008	0.006	0.009	0.00E+00
I10	K70	0.003	0.002	0.004	0.00E+00
I10	C43	0.001	0.000	0.002	2.56E-03
I10	I77	0.006	0.005	0.007	0.00E+00
I10	K85	0.003	0.003	0.004	0.00E+00
I10	L97	0.006	0.005	0.007	0.00E+00
I10	G93	0.006	0.004	0.007	0.00E+00
I10	M21	0.003	0.003	0.004	0.00E+00
I10	K91	0.003	0.002	0.004	0.00E+00
I10	I46	0.006	0.005	0.007	0.00E+00
I10	J69	0.006	0.005	0.008	0.00E+00
I10	E16	0.006	0.004	0.008	0.00E+00
I10	H43	0.003	0.002	0.004	0.00E+00
I10	K50	0.001	0.000	0.001	2.46E-02
I10	I31	0.007	0.006	0.008	0.00E+00
I10	C15	0.001	0.000	0.001	3.63E-02
I10	J38	0.002	0.001	0.003	1.00E-07
I10	B97	0.008	0.007	0.009	0.00E+00
I10	I65	0.008	0.007	0.009	0.00E+00
I10	F03	0.005	0.004	0.006	0.00E+00
I10	M50	0.002	0.002	0.003	0.00E+00
I10	L08	0.003	0.002	0.004	0.00E+00
I10	N23	0.000	0.000	0.001	2.08E-01
I10	C20	0.002	0.001	0.002	1.08E-04

I10	H54	0.004	0.003	0.005	0.00E+00
I10	K04	0.002	0.001	0.002	6.00E-05
I10	I27	0.005	0.004	0.005	0.00E+00
I10	H18	0.002	0.001	0.003	1.00E-07
I10	G30	0.002	0.002	0.003	0.00E+00
I10	G31	0.004	0.003	0.005	0.00E+00
I10	M46	0.003	0.002	0.004	0.00E+00
I10	N19	0.003	0.002	0.004	2.00E-07
I10	H90	0.003	0.002	0.004	0.00E+00
I10	E53	0.005	0.004	0.006	0.00E+00
I10	J12	0.007	0.006	0.008	0.00E+00
I10	N45	0.002	0.001	0.003	0.00E+00
I10	K61	0.001	0.000	0.001	5.95E-02
I10	B98	0.002	0.001	0.003	4.47E-05
I10	K55	0.003	0.002	0.004	0.00E+00
I10	U82	0.006	0.005	0.007	0.00E+00
I10	C80	0.001	0.000	0.002	7.37E-02
I10	C85	0.000	0.000	0.001	1.44E-01
I10	I74	0.011	0.010	0.012	0.00E+00
I10	I12	0.002	0.001	0.002	0.00E+00
I10	N21	0.002	0.001	0.003	0.00E+00
I10	H00	-0.001	-0.001	0.000	2.59E-02
I10	M70	0.001	0.001	0.002	1.09E-04
I10	K74	0.003	0.002	0.004	0.00E+00
I10	C64	0.003	0.003	0.004	0.00E+00
I10	G99	0.003	0.002	0.003	0.00E+00
I10	A49	0.004	0.003	0.004	0.00E+00
I10	B34	0.002	0.001	0.003	3.00E-06
I10	J93	0.003	0.002	0.004	0.00E+00
I10	K90	0.001	0.000	0.002	1.98E-02
I10	D37	0.002	0.001	0.003	3.00E-07
I10	C25	0.001	0.000	0.001	1.47E-01
I10	D18	0.001	0.001	0.002	1.31E-04
I10	M84	0.001	0.001	0.002	1.68E-04
I10	K82	0.003	0.002	0.004	0.00E+00
I10	I61	0.005	0.004	0.006	0.00E+00
I10	F00	0.002	0.001	0.003	0.00E+00
I10	M62	0.002	0.001	0.003	1.00E-07
I10	C16	0.001	0.000	0.001	1.29E-02
I10	C83	0.000	0.000	0.001	1.59E-01
I10	D47	0.003	0.003	0.004	0.00E+00
I10	H81	0.003	0.002	0.004	0.00E+00

I10	M89	0.001	0.001	0.002	1.13E-05
I10	G25	0.003	0.002	0.003	0.00E+00
I10	K01	0.000	0.000	0.001	4.06E-01
I10	M77	0.001	0.000	0.001	9.33E-03
I10	I64	0.002	0.001	0.003	7.50E-06
I10	M86	0.002	-0.006	0.010	6.91E-01
I10	M45	0.001	0.000	0.001	3.00E-03
I10	F31	0.001	0.000	0.001	9.06E-03
I10	N31	0.000	0.000	0.000	7.75E-01
I10	K14	0.001	0.001	0.002	5.92E-05
I10	D41	0.002	0.001	0.002	1.00E-06
I10	K65	0.002	0.001	0.002	3.46E-05
I10	D04	0.000	0.000	0.001	1.92E-01
I10	K12	0.002	0.001	0.003	2.40E-06
I10	M41	0.002	0.001	0.003	0.00E+00
I10	M07	0.001	0.001	0.002	4.90E-06
I10	D68	0.001	0.001	0.002	1.63E-05
I10	F01	0.003	0.002	0.004	0.00E+00
I10	G63	0.023	0.020	0.025	0.00E+00
I10	C71	0.001	0.000	0.001	1.61E-02
I10	C90	0.001	0.000	0.001	7.48E-02
I10	F20	0.000	0.000	0.001	1.86E-01
I10	C91	0.000	0.000	0.001	3.32E-01
I10	U83	0.004	0.003	0.004	0.00E+00
I10	L60	0.001	0.000	0.001	1.98E-02
I10	H65	0.000	0.000	0.001	3.73E-01
I10	D13	0.001	0.000	0.001	1.02E-04
I10	H93	0.003	0.002	0.003	0.00E+00
I10	C19	0.001	0.001	0.002	8.94E-05
I10	G51	0.002	0.002	0.003	0.00E+00
I10	I72	0.004	0.003	0.004	0.00E+00
I10	H61	0.000	0.000	0.001	1.95E-01
I10	J31	0.001	0.000	0.001	2.80E-03
I10	K75	0.002	0.002	0.003	0.00E+00
I10	D75	0.002	0.001	0.002	3.10E-06
I10	A08	0.002	0.001	0.003	0.00E+00
I10	D63	0.002	0.002	0.003	0.00E+00
I10	H34	0.003	0.002	0.003	0.00E+00
I10	K11	0.001	0.000	0.001	5.16E-02
I10	G95	0.001	0.001	0.002	1.50E-05
I10	H11	0.001	0.000	0.002	5.02E-04
I10	I22	0.001	0.000	0.001	8.97E-04

I10	D03	0.001	0.000	0.001	1.91E-02
I10	J15	0.002	0.001	0.003	6.09E-05
I10	I85	0.001	0.001	0.002	1.05E-04
I10	I89	0.002	0.002	0.003	0.00E+00
I10	C97	0.002	0.001	0.003	0.00E+00
I10	M71	0.001	0.001	0.002	7.40E-06
I10	D35	0.002	0.002	0.003	0.00E+00
I10	D86	0.001	0.000	0.001	5.41E-03
I10	D61	0.001	0.001	0.002	2.82E-05
I10	I87	0.002	0.001	0.003	0.00E+00
I10	C22	0.002	0.001	0.002	0.00E+00
I10	E27	0.002	0.001	0.003	0.00E+00

3. Impact of Treatment and Healthcare Access: The study's analysis of disease progression patterns does not appear to account for the variability in treatment efficacy or access to healthcare services, which can significantly influence the course of multimorbidity. Differences in healthcare access and treatment responses could confound the relationships between diseases. Yet, there is no discussion on how these factors were considered in the analysis or how they might affect the interpretation of the results. Including such considerations is crucial for understanding the real-world applicability of the findings.

Response: We apologize for the lack of clarity. We fully agree with the Reviewer that including confounders in relation to health access and treatment is crucial, and we endeavored to include possible proxy variables from the UK Biobank dataset. Specifically, we included all disease status and factors influencing health status and contact with health services (ICD codes, Z00–Z99). Factors influencing health status and contact with health services include ICD-10 codes for Persons encountering health services for examinations (Z00–Z13), Genetic carrier and genetic susceptibility to disease (Z14–Z15), Resistance to antimicrobial drugs (Z16–Z16), Estrogen receptor status (Z17–Z17), Retained foreign body fragments (Z18–Z18), Hormone sensitivity malignancy status (Z19–Z19), Persons with potential health hazards related to communicable diseases (Z20–Z29), socioeconomic and psychosocial circumstances (Z55–Z65), family and personal history and certain conditions influencing health status (Z77–Z99), Persons encountering health services in circumstances related to reproduction (Z30–Z39), Encounters for other specific health care (Z40–Z53), Do not resuscitate status (Z66–Z66), Blood type (Z67–Z67), Body mass index (Z68–Z68), Persons encountering health services in other circumstances (Z69–Z69). These variables indicate a reason for an encounter or provide additional information about a patient encounter.

Nevertheless, we acknowledged the current treatment practices may not be fully captured by the ICD-10 code data and have added texts to strengthen the discussion on this limitation as the Reviewer suggested:

Lines 129-132: “Additionally, all disease status and factors influencing health status and contact with health services (ICD codes, Z00–Z99) at baseline were included to control confounding (appendix p6). These variables indicate a reason for an encounter or provide additional information about a patient encounter.”

Discussions Lines 479-484: “Furthermore, our data does not account for the contributions of biological mechanisms and current treatment practices to the causal links among diseases. To conduct such causal and mechanistic investigations, a data-rich and thoroughly phenotyped dataset with treatment details is required. Nevertheless, our findings generate hypotheses that may be tested in a smaller, well-characterized sample.”

4. Methodological Clarity and Reproducibility: The manuscript overviews the analytical methods but lacks detailed explanations of the machine learning algorithms used, the rationale for their selection, and the parameter settings. For such a complex analysis involving machine learning and TMLE, it is essential to provide sufficient methodological detail to ensure the reproducibility of the findings. Furthermore, the article mentions using an appendix for supplementary analyses but does not integrate these supplementary results effectively into the discussion of the main findings.

Response: As suggested, we have added the following paragraph to clarify the details of TMLE and incorporated the alternative sensitivity analysis on stability into the main findings:

Appendix Lines 141-153: “Specifically, it was conducted following the four steps:
Step 1: We estimated the expected models using the logistic models, with all baseline covariates and exposure disease status as independent variables.
Step 2: We estimated the propensity scores models using the logistic models, with all baseline covariates status as independent variables, which were then used to build the fluctuation parameters, which are the inverse probability of exposure minus the inverse probability of unexposure.
Step 3: We updated the expected outcome models from Step 1 by fitting a logistic regression using the fluctuation parameters estimated from Step 2 as the only predictor and the initially expected outcome under the observed exposure as a fixed intercept.
Step 4: We calculated the average difference between the updated expected outcomes from Step 3, under the two exposure conditions.”

Lines 296-298: “Higher adjusted Rand indexed indicate greater clustering stability. Adjusted Rand Indexes from sensitivity analyses were still sufficiently high, 0.65 and 0.68 for clusterings in females and males, respectively.”

For the clustering algorithms, we provided the details of parameter settings in supplementary materials (lines 265-269).

5. Generalizability of Findings: The study is based on UK Biobank data, which, while extensive, may not fully represent the global diversity in disease progression patterns due to geographic, ethnic, and socioeconomic variations. The manuscript briefly mentions this in the limitations but does not critically evaluate the potential biases introduced by relying on a single-country dataset. A more thorough discussion of how these limitations might affect the applicability of the findings to other populations and settings would make the limitations section more robust.

Response: The Reviewer brought up a valid concern about the applicability of our findings to other countries. While it is true that our dataset only includes data from a single country, we believe that our findings could still be relevant to other countries. This is because we have accounted for many variables

that could have confounded our results, such as socioeconomic factors (including age, sex, race, education level, deprivation, and social activities), health behaviors (such as smoking, alcohol use, sleep duration, phone use, physical activity, and gas use), health status, doctor-diagnosed diseases, family history, and all disease status and factors influencing health status and contact with health services.

However, it is possible that there are other confounding variables that we did not account for in our dataset. These variables could include treatment practices, environmental factors, genetics, dietary habits, and ethnic and socioeconomic factors that were not captured in our data. Therefore, we cannot say for certain whether our findings would be applicable to countries that have large differences in these factors compared to the United Kingdom. Further investigation is needed to determine the generalizability of our findings to other countries. This has been clarified in the revised text:

Lines 484-489: “Finally, while our dataset is limited to a single country, we believe that our findings could still apply to other countries. We have controlled for many variables, such as socioeconomic factors, health behaviors, health status, and more. However, there may still be other confounding variables that we did not account for, such as treatment practices, environmental factors, genetics, and dietary habits. Further investigation is needed to determine the generalizability of our findings to other countries.”

6. Validation of Clustering Methods: While the manuscript details the use of self-tuning k-means clustering and Kamada-Kawai visualization for identifying multimorbidity progress constellations, there is insufficient discussion on validating these clustering results. Given the potential for clustering algorithms to yield varying results depending on parameter settings and initial conditions, a rigorous validation approach is necessary, possibly including stability analysis or comparison with other clustering methods. Such validation is crucial for ensuring the reproducibility and generalizability of the identified multimorbidity patterns.

Response: We apologize for this lack of clarity. We assessed the stability of clustering in two different analyses. In the former stability analysis, we compared our clusterings with clusterings from traditional k-means method with different initial random seeds. In the alternative sensitivity analysis, we tested the stability by comparing results across distinct clustering methods. In the original submission, we provided the details in Supplementary Materials (lines 220-283, p10, Supplementary Materials). In this revision, we have added it to the text to clarify this:

Lines 158-161: “We tested the stability of clustering by comparing our clusterings with clusterings from traditional k-means method with different initial random seeds, and self-tuning spectral clustering method. As an alternative sensitivity analysis, we tested the stability by comparing results across distinct clustering methods (appendix p 13).”

The results were reported in the text as follows:

Lines 294-298: “We identified 10 constellations for females (clustering stability, mean adjusted Rand index 0.76, figure 5) and 9 constellations for males (clustering stability, mean adjusted Rand index 0.75, figure 6). Higher adjusted Rand index indicate greater clustering stability. Adjusted Rand Indexes from sensitivity analyses were still sufficiently high, 0.65 and 0.68 for clusterings in females and males respectively.”

7. Handling of Sex-Specific Differences: The study highlights interesting sex-specific differences in

multimorbidity patterns but does not sufficiently explore the biological or sociodemographic factors that might explain these differences. A more critical analysis of whether these differences could be attributed to methodological biases, differential healthcare utilization, or genuine biological distinctions would provide deeper insights into the observed patterns.

Response: Thank you, this has been clarified in the revised text:

Discussions Lines 453-455: “Note that we used the same methods for females and males and have controlled the sociodemographic factors and healthcare utilization, the findings on sex-specific differences are more likely to represent the biological differences.”

8. Assessment of Disease Classification System: The manuscript utilizes the International Classification of Diseases (ICD-10) for categorizing diseases, which is a standard approach. However, the study does not critically assess the limitations of this classification system in capturing the nuances of multimorbidity progression, especially for diseases that might straddle multiple categories or for emerging diseases not well-represented in ICD-10. A critical discussion on the implications of these classification challenges for the study's findings would enhance the manuscript's depth.

Response: As suggested, we have added the text in the Discussions as follows:

Discussions Lines 495- 505: “As a statistical classification, the ICD-10 code has linearization properties, notably mutual exclusivity of categories and exhaustive coverage of the domain of interest. However, disease conditions are often complex and can involve multiple parts of the body. As a result, some conditions in the ICD-10 code may not be located in appropriate chapters, which can lead to incorrect counting of multimorbidity progress within and across chapter-level categories. In fact, our identified patterns of cross-chapter multimorbidity progress could help investigate such potentially inappropriate classification, where higher clustering cross-chapter tendency may indicate relationships of certain diseases to both chapters. Furthermore, the ICD-10 code may not accurately capture emerging diseases, which are classified under the new diseases of uncertain etiology or emergency use chapter (U00-U49). However, these emerging diseases are typically less prevalent and thus are not the focus of the study.”

We would like to note that all the prevalence of the ICD codes in the new diseases of uncertain etiology or emergency use chapter (U00-U49) were less than 1%, except for emergency use (U07), which was prevalent among males only.

D. Minor Comments

1. **Typographical Errors:** The manuscript contains minor typographical errors (e.g., inconsistencies in disease names and slight grammatical errors) that should be corrected for readability.

Response: Thank you. We have reviewed and corrected the ICD-10-chapter names and disease names. In particular, we have revised the “diseases” in chapter names by “chapter” to avoid confusion in the text. Note that we have ensured the original names of the ICD-10 code per codebook in the figures while we slightly simplify the names in the main text for readability, as in the literature (Jensen et al. 2014; Westergaard et al. 2019). Specifically, we remove the “other” from the beginning of disease names in the text, such as "other chronic obstructive pulmonary disease (J44)" is now "chronic obstructive pulmonary

disease (J44)". Additionally, we have simplified "essential (primary) hypertension (I10)" to "hypertension (I10)".

2. **Figure and Table Clarity:** While the figures and tables are generally informative, some could benefit from higher resolution or clearer labeling to enhance their interpretability.

Response: Thank you. We have independently submitted high-resolution figures in this revision.

Also, we have added disease names for the figures in the main text. Specifically, we have added the disease names in Figure 3 and the disease names of chapters in Figure 4. We checked all other relevant figures in the main text (Figures 2, 5, 6), for which we have provided disease names.

Reviewer #2 (Remarks to the Author):

Thank you for the opportunity to review the manuscript by Dr. Han et al titled "Mapping multimorbidity progression among 190 diseases"

In this work, the authors used the rich UK BioBank data set (500,000 participants 37-73 years, mean 57 and mean follow-up of 12.7 years).

The study aims to describe the progression of multimorbidity and map clusters of this progression; this entailed a complex matrix of correlations and other analysis in a pairwise fashion also to identify "influential" and "influenced: diseases as central point driving multimorbidity and multimorbidity pathways. One of the biggest claims of this study compared to previous publications is that Dr. Han et al provide their models' causal relationship rather than just association relationship.

Response: The Reviewer's keen attention to detail and grasp of our main contributions is truly impressive and appreciated. We would like to thank him/her for taking the time to understand and acknowledge our efforts.

However, we wish to clarify a minor point: we did not specifically state our intention to describe the progression of multimorbidity. We respectfully offer this clarification, as the term "describe" could potentially cause confusion with describing life-course multimorbidity by describing the number and the type of diseases each individual has experienced since childhood or infancy, which we believe may be pertinent to the Reviewer's second comment.

Specifically, we clearly stated in Abstract that "we estimated the causal relationships among prevalent diseases and mapped out the clusters of multimorbidity progression among them [lines 40-41]", in Main section "we address this gap by providing a framework to examine the progression of multimorbidity and map the clustering of progression based on causal relationships between diseases [Lines 103-104] ", and in Discussion section "in this study, we examine how multimorbidity progresses among 190 diseases by examining the comprehensive causal links among them and analyzing their coalescing patterns [Lines 371-372]".

The idea and concept are quite attractive: discover the natural progression of diseases in the context of multimorbidity, identify the influencer, direct attention to it, and prevent progression to more complex clusters of irreversible diseases. This is the holy grail of multimorbidity prevention and public health. The number of participants is large.

Response: We want to express our gratitude for recognizing the potential impact of this work. Your positive feedback on the idea and concept, which you described as "quite attractive," is deeply appreciated. Your acknowledgment of this as the "holy grail" in the field of multimorbidity prevention and public health is incredibly inspiring.

Major comments and overall impression:

1. The analysis and report are extremely complex with pretty illustrations but missing practical (clinical) interpretation. This is the most salient drawback, the project is ambitious and mathematically correct, however how as a clinician, health care policy maker, researcher, etc translate this in practical terms? This comment relates to several observations:

Response: We appreciate the valuable input from Reviewer 2 who has provided insightful observations on our analysis and report. It is encouraging to hear that our work has been recognized for its potential impact and mathematical accuracy, as well as the visually appealing illustrations.

We would like to respectfully acknowledge the concern raised by the reviewer regarding the lack of clinical interpretation in our report. However, we would like to clarify that we have addressed this issue in our Discussion section. In particular, we provided a comprehensive discussion of the clinical implications of the identified top influential and influenced diseases (lines 382-399), ray clusterings and ring clusterings (lines 400-411), bi-directional and uni-directional one-step progression cross chapters (lines 412-422), multimorbidity progress constellations (lines 423-451), and sex differences in patterns (lines 452-462), and potential mechanisms underlying the development of multimorbidity (lines 463-472).

However, after carefully reviewing the reviewer's specific feedback following this comment, we have identified areas where we could have provided more detail on the clinical implications of the most influential and influenced diseases. We have taken this feedback on board and made the necessary revisions to our report to ensure that it meets the highest standards of clarity and practical relevance. Please refer to our detailed response below, including our responses to comments #1.a, #1.b, and the last major comment.

a. It is quite ambitious to map all disease trajectories, therefore you ended up with ICD-10 codes that have little meaning to the reader. ICD-10 codes sometimes have funny interpretations and ambiguous meanings to the clinician.

Response: First of all, we chose to utilize the ICD-10 codes for our analysis for three primary reasons: **1)** this approach allows for easy retrieval and analysis of data by translating diagnoses and other health problems from words into alphanumeric codes (World Health Organisation International Classification of Diseases and Related Health Problems' ICD-10 Volume 2, 2.1 Purpose and applicability). **2)** The use of ICD-10 codes can foster consistency and comparability of the data at local, national and global levels, which helps establish our study as a basis to support countrywide comparison in the future. It is documented that "the classification of diagnoses using ICD-10 is a mandatory national requirement", and "National Health Service (NHS) managers and health care professionals use it locally to support operational/strategic planning and performance management"(National Clinical Coding Standards ICD-10 5th Edition 2023 [V9.0]. pp3-6). Furthermore, the United Kingdom and many other countries have a mandatory obligation to collect and submit ICD-10 data to the World Health Organisation (WHO) to facilitate the production of international statistical and epidemiological data (SCCI0021: International Statistical Classification of Diseases and Health Related Problems [ICD-10] 5th Edition, WHO Nomenclature Regulations adopted by the World Health Assembly in 1967). **3)** The list of illnesses consists of ailments that share the same level of categorization (ICD-10 at the third tier). This ensures a uniform definition of multimorbidity patterns. We believe that in multimorbidity studies, it is crucial to utilize uniformly clinically assessed conditions and validated disease lists to guarantee consistency and generalizability across research studies (Davide Liborio Vetrano et al. 2021).

We understand the reviewer's concern about the potential ambiguity on interpretation and implementation of the ICD-10 codes, when lacking clear directives, may jeopardize data consistency and comparability in relation to the classification of ICD-10 codes. Nonetheless, WHO and national health departments have implemented specific guidelines through national clinical coding standards to ensure data consistency in areas that may present ambiguity. Such instructions are comprehensively outlined in

the National Clinical Coding Standards ICD-10 5th Edition 2023 in the United Kingdom, to the fullest extent feasible.

Nevertheless, we agree with the reviewer that the ICD-10 code may appear too symbolic, so we attempted to provide both the code and the disease name in the original submission. In this revision, we have carefully checked the text to ensure that both the ICD-10 code and disease names are appropriately and consistently provided.

b. The graphs are quite beautiful but with little meaning.

Response: We thank the reviewer for the kind word on considering our graphs are “quite beautiful”. We have made a great effort to plot these figures to ensure the informative. The reviewer’s criticism of meanings is too broad and too vague for us to apprehend. We believe a bit of clear specificity would help us to provide a cogent response. However, we shall understand this as 1) a lack of referring to the figures when discussing the clinical implications in the text and 2) a lack of disease names in figures, as pointed out by the reviewer in the last comment.

For 1), we have checked the full text to ensure the figures are referred when discussing the relevant findings.

For 2), we have checked all six figures and figure legends in the main text to ensure all disease names are provided in the figures. Specifically,

- we have added the disease name for each ICD-10 code in Figure 3 and the disease names of chapters in Figure 4.
- we checked all other relevant figures in the main text (Figures 2, 5, 6), for which we have provided disease names in the original submission.
- we note that Figure 1 is an illustration of the framework of the study, not pertinent to the comments.

As for the figures in the Supplementary materials, we have chosen different strategies to provide the disease names based on the purpose of the graphical illustrations. Specifically,

- we have provided diseases for each ICD-10 code in Extended Data Figs. 6 and 7, and Supplementary Figs. 2 in the original submission.
- for Extended Data Figs. 1 and 2, we provided ways how to obtain disease names in the figure legends. The two figures aim to illustrate the clustering pattern of diseases based on how they affect other diseases, and alternatively, the clustering pattern of diseases based on how other diseases affect them, as well as how the identified top diseases in Figure 3 clustered. The current illustrations would give clear patterns while adding names in the plots will make the patterns confusing and less apparent.
- we keep Extended Data Figs. 3-5 unchanged. Extended Data Figs. 3 is a methodological illustration provided to show the stability of clustering. Extended Data Figs. 4 and 5, illustrate the causal effects for the top 100 one-step multimorbidity progress. The focus of the graphical illustration is on the statistical values.
- we note that Supplementary Figs.1 does not involve ICD10-codes and is not pertinent to the comments.

2. If disease trajectories are the central point of your study, the data set you are using and the follow-up time and time to the association are not the appropriate ones:

a. The average age is 57 years in your cohort: in the UK > 60% of your cohort have at least one comorbidity. Therefore, if you want to establish trajectories for chronic diseases you must study a pediatric population followed through the life course, which is not the case here.

Response: We would like to clarify respectfully that we did not aim to describe the life-course trajectories of multimorbidity by deriving the number and type of diseases each individual has experienced since childhood or infancy from the real data. It is true that the current data might be inappropriate for such studies. However, this is not the purpose of the current study, which clearly states that we aim to estimate the causal relationships among prevalent diseases and map out the clusters of multimorbidity progression among them [lines 40-41].

As the reviewer cleverly pointed out, our study “compared to previous publications provide[s]...causal relationship rather than just association relationship.” We would like to note that the previous publications, such as Jensen et al. (2014), Westergaard et al. (2019), Vetrano et al. (2020), and Dervic et al. (2024), all did not follow a pediatric population through the life course. Moreover, compared to the data used for association analysis in these previous studies, the UK Biobank dataset is a “rich dataset”. With such a rich dataset, we can include many informative covariates as confounders to increase the plausibility of the unmeasured unconfoundedness assumption in causal inference (Rosenbaum 2002, Rubin 2009).

b. Many chronic diseases initiate at an early age, have a long latency period and manifest at mid and old age.

Response: The reviewer’s comments are a bit vague for us to apprehend. As the reviewer quickly pointed out, we used the rich UK BioBank data set (500,000 participants 37-73 years, mean 57, and mean follow-up of 12.7 years). The length of follow-up is close to the existing literature. For example, Jensen et al. 2014 used data from the Danish National Patient Registry (NPR), which covers all hospital encounters for a 14.9-year period from 1996 to 2010.

Alternatively, if the reviewer’s point is conducting a long-term pediatric cohort study, please see our previous response to #Comment 1. a.

c. Diseases with higher prevalence and easier to measure (blood pressure is easier than having a chest CT to detect lung cancer) will skew any multimorbidity analysis; in the your methods a thorough evaluation is performed on participants at enrollment, but encounters with the health systems obtain the follow-ups: the latter creates the above bias= at every encounter we will measure the blood pressure, therefore increasing the chance to diagnose hypertension, but less likely to detect diseases that need sophisticated studies: echo, chest CT, dexa-scans, blood markers etc. Therefore, the trajectory described reflects how UK physicians practice and order tests rather than a natural progression of diseases. To address this issue, you must measure a comprehensive battery of tests and questionnaires at established interval times to detect and establish the true prevalence.

Response: We would like to express our gratitude to the reviewer for bringing up the insightful point. We concur with the reviewer's observation that this population-based cohort study may be biased by measurement errors, particularly if these errors are systematic or differential depending on the earlier disease status.

We clarify that our findings can only represent the observed disease progression as it is subject to the current clinical practice in the real world. We acknowledged this relevance in the original submission. Upon conducting a secondary analysis, we discovered that the majority of diseases (86.9% for females and 89.1% for males) displayed an elevated causal effect within the first year of follow-up (appendix pp 29-31). This could be due to the fact that illnesses that manifest after the initial diagnosis are more likely to be detected and diagnosed within a shorter period than over an extended period. However, whether this reflects the natural progression of multimorbidity or if screening strategies require further investigation (lines 392-399).

However, it's also worth noting that it is practically infeasible to repeatedly administer all diagnostic tests for the 190 diseases to each individual, due to ethical concerns (certain screenings are body-intrusive) and financial constraints. Nevertheless, our team, led by the corresponding author Chen Wang, is currently conducting a comparable cohort study in China that involves repeated and thorough evaluations of individuals during follow-ups (Ref 60), though with a shorter disease list for practical concerns. We fully agree with the reviewer that gathering more comprehensive follow-up data is crucial in addressing this matter. We hope this can be accomplished within the next decade. In this revised version, we have explicitly stated this limitation:

Discussions Lines 490-494: "Also, it's possible that the study contains inaccuracies in measurements, such as certain conditions that are presently undetectable or undiagnosed in clinical settings, which could lead to biased estimates. To address this challenge, it's crucial to gather reliable data from population-based cohort studies that conduct comprehensive and regular assessments of individual disease status throughout the duration of the study."

d. to establish a disease leading to another disease a period of one year is not adequate.

Response: In our **primary analysis**, we used the complete dataset of UK Biobank, with an average follow-up period of 12.7 years. However, we conducted a **sensitivity analysis**, limiting our analysis to a one-year follow-up period and repeating the causal analysis to assess the rapidity of disease impact.

Our findings showed that a majority of diseases exhibited an elevated causal effect within the first year of follow-up. This could be due to conditions that manifest after the initial diagnosis are more likely to be detected and diagnosed within a shorter period than over an extended period. It remains unclear whether this reflects the natural progression of multimorbidity or if screening strategies require further investigation. This finding also reflects our efforts to understand the potential impact of measurement bias, which is relevant to Comment #1.c.

To illustrate my comments, on page 11 lines 204-211 of the main manuscript, the authors describe two entities that "prevent" disease progression. as the authors wrote, can help with "multimorbidity prevention": should I recommend my patients to wear glasses and have haemorrhoids? This paragraph summarizes the mathematical-to-clinical disconnect that I described above.

Response: Thank you for this illustration. It really helps us pinpoint your concerns about the mathematical-to-clinical disconnect. In the paragraph, we reported the findings on the top 10 diseases that are less likely to be associated with progression to multimorbidity (with disorders of refraction and accommodation as the top 1) and the top 10 diseases that are less likely to develop after initially having other diseases (with haemorrhoids as the top 1). As our response to #Comment 1.c, the analysis could only reflect the progression of disease in the real world with current clinical practice. We have revised the text in several places to clarify this.

Discussions Lines 383-391: “The profiles help healthcare providers in recognizing the risk of subsequent diseases while assessing patients without multimorbidity, and in determining intervention and prevention strategies to prevent multiple long term conditions. We emphasized this profile should be cautiously interpreted with data limitations we shall discuss later. Whether the identified causal relationships between two diseases are a result of the natural development mechanism of multimorbidity, clinical treatment practice for the initial diagnosis, patients’ adapting to healthier lifestyles after the initial diagnosis, or if screening strategies require further investigation.”

Minor issues:

Please provide a table 1

Response: Thank you. Supplementary Table 1 was described in Supplementary Materials (lines 134-135). We have amended the main text to refer to it.

In your graphs substitute ICD-10 codes and use common diseases names for ease of interpretation.

Response: As per your suggestion, we have updated all the figures in the main text to now include disease names. To further enhance clarity and enable easy identification of specific diseases, we have also kept the ICD-10 codes. With a total of 190 diseases being studied, these codes will prove to be a valuable tool for grasping the key information of the figures and facilitating comprehension across borders. It's worth noting that these codes have been translated into 43 languages and are widely used by member states in accordance with the WHO Nomenclature Regulations established in 1967.

References

Rosenbaum, P. (2002). *Observational Studies*, Springer.

Rubin, D. B. (2009). Should observational studies be designed to allow lack of balance in covariate distributions across treatment groups? *Statistics in Medicine* 28, 1420-1423.

Imbens GW, Rubin DB. *Causal Inference for Statistics, Social, and Biomedical Sciences: An Introduction*. Cambridge University Press; 2015.

Stürmer T, Webster-Clark M, Lund JL, et al. Propensity score weighting and trimming strategies for reducing variance and bias of treatment effect estimates: A simulation study. *American Journal of Epidemiology*. 2021;190(8):1659-1670.

World Health Organization. (2015). International statistical classification of diseases and related health problems, 10th revision, Fifth edition, 2016. World Health Organization. <https://iris.who.int/handle/10665/246208>

National Clinical Coding Standards ICD-10 5th Edition (2023). NHS England. https://classbrowser.nhs.uk/ref_books/ICD-10_2023_5th_Ed_NCCS.pdf

SCCI0021: International Statistical Classification of Diseases and Health Related Problems (ICD-10) 5th Edition.[Internet] 2015 Available from: <http://www.digital.nhs.uk/isce/publication/scci0021>.

WHO Nomenclature Regulations adopted by the World Health Assembly in 1967. Accessed 20 April 2024. <https://treaties.un.org/doc/publication/unts/volume%201172/volume-1172-i-18749-english.pdf>.

Vetrano, D.L., Dekhtyar, S. and Triolo, F., (2021). Mens sana in corpore sano: multimorbidity and mental health. The Lancet Regional Health–Europe, 8.

Vetrano, D.L., Roso-Llorach, A., Fernández, S. et al. (2020). Twelve-year clinical trajectories of multimorbidity in a population of older adults. Nature Communications,11, 3223.

Jensen AB, Moseley PL, Oprea TI, et al. (2014). Temporal disease trajectories condensed from population-wide registry data covering 6.2 million patients. Nature Communications, 5, 4022.

Westergaard, D., Moseley, P., Sørup, F. K. H., Baldi, P., & Brunak, S. (2019). Population-wide analysis of differences in disease progression patterns in men and women. Nature Communications, 10, 666.

Dervić, E., Sorger, J., Yang, L. et al. (2024). Unraveling cradle-to-grave disease trajectories from multilayer comorbidity networks. npj Digital Medicine, 7, 56.

Appendix Table: Disease names for the 100 less prevalent diseases.

ICD-10	Disease names
G20	Parkinson's disease
K35	Acute appendicitis
J30	Vasomotor and allergic rhinitis
K70	Alcoholic liver disease
M67	Other disorders of synovium and tendon
C43	Malignant melanoma of skin
I77	Other disorders of arteries and arterioles
K85	Acute pancreatitis
L97	Ulcer of lower limb, not elsewhere classified
G93	Other disorders of brain
M21	Other acquired deformities of limbs
K91	Postprocedural disorders of digestive system, not elsewhere classified
I46	Cardiac arrest
J69	Pneumonitis due to solids and liquids
H04	Disorders of lachrymal system
E16	Other disorders of pancreatic internal secretion
H43	Disorders of vitreous body
K50	Crohn's disease [regional enteritis]

I31	Other diseases of pericardium
C15	Malignant neoplasm of oesophagus
J38	Diseases of vocal cords and larynx, not elsewhere classified
B97	Viral agents as the cause of diseases classified to other chapters
I65	Occlusion and stenosis of precerebral arteries, not resulting in cerebral infarction
F03	Unspecified dementia
M50	Cervical disk disorders
E55	Vitamin D deficiency
L08	Other local infections of skin and subcutaneous tissue
N23	Unspecified renal colic
G43	Migraine
C20	Malignant neoplasm of rectum
H54	Blindness and low vision
M35	Other systemic involvement of connective tissue
M43	Other deforming dorsopathies
K04	Diseases of pulp and periapical tissues
I27	Other pulmonary heart diseases
H18	Other disorders of cornea
G30	Alzheimer's disease
G31	Other degenerative diseases of nervous system, not elsewhere classified
M46	Other inflammatory spondylopathies
N19	Unspecified renal failure
R15	Faecal incontinence
H90	Conductive and sensorineural hearing loss
E53	Deficiency of other B group vitamins
J12	Viral pneumonia, not elsewhere classified
N45	Orchitis and epididymitis
K61	Abscess of anal and rectal regions
B98	Other specified infectious agents as the cause of diseases classified to other chapters
K55	Vascular disorders of intestine
L90	Atrophic disorders of skin
U82	Resistance to betalactam antibiotics
C80	Malignant neoplasm without specification of site
C85	Other and unspecified types of non-Hodgkin's lymphoma
I74	Arterial embolism and thrombosis
I12	Hypertensive renal disease
N21	Calculus of lower urinary tract
H00	Hordeolum and chalazion
M70	Soft tissue disorders related to use, overuse and pressure
K74	Fibrosis and cirrhosis of liver
C64	Malignant neoplasm of kidney, except renal pelvis
G99	Other disorders of nervous system in diseases classified elsewhere

A49	Bacterial infection of unspecified site
B34	Viral infection of unspecified site
J93	Pneumothorax
K90	Intestinal malabsorption
D37	Neoplasm of uncertain or unknown behaviour of oral cavity and digestive organs
C25	Malignant neoplasm of pancreas
D18	Haemangioma and lymphangioma, any site
M84	Disorders of continuity of bone
K82	Other diseases of gallbladder
I61	Intracerebral haemorrhage
F00	Dementia in Alzheimer's disease
M62	Other disorders of muscle
C16	Malignant neoplasm of stomach
C83	Diffuse non-Hodgkin's lymphoma
D47	Other neoplasms of uncertain or unknown behaviour of lymphoid, haematopoietic and related tissue
H81	Disorders of vestibular function
M85	Other disorders of bone density and structure
M89	Other disorders of bone
G25	Other extrapyramidal and movement disorders
K01	Embedded and impacted teeth
M77	Other enthesopathies
I64	Stroke, not specified as haemorrhage or infarction
M86	Osteomyelitis
M45	Ankylosing spondylitis
F31	Bipolar affective disorder
N31	Neuromuscular dysfunction of bladder, not elsewhere classified
K14	Diseases of tongue
D41	Neoplasm of uncertain or unknown behaviour of urinary organs
K65	Peritonitis
D04	Carcinoma in situ of skin
K12	Stomatitis and related lesions
E05	Thyrotoxicosis [hyperthyroidism]
M41	Scoliosis
M07	Psoriatic and enteropathic arthropathies
D68	Other coagulation defects
F01	Vascular dementia
G63	Polyneuropathy in diseases classified elsewhere
C71	Malignant neoplasm of brain
C90	Multiple myeloma and malignant plasma cell neoplasms
F20	Schizophrenia

REVIEWERS' COMMENTS:

Reviewer #1 (Remarks to the Author):

The authors have effectively addressed most of my major comments, providing detailed explanations and justifications for their methods and the limitations of their study. Their responses demonstrate careful consideration of my review comments and a satisfactory effort to improve the manuscript's clarity, robustness, and generalizability.

There are areas, particularly in "discussing the impact of excluding rarer diseases" and "further exploring the generalizability of the findings," where the revision could provide even greater depth and promise for the study's broader applicability.

Overall, the revisions and responses indicate significant improvements to the manuscript. Therefore, I am pleased to accept the article as is or after addressing my minor comment above. I leave this decision to the editor.

Reviewer #2 (Remarks to the Author):

Thank you for your thorough response to my comments.

Responses to Reviewers' specific comments

Reviewer #1 (Remarks to the Author):

The authors have effectively addressed most of my major comments, providing detailed explanations and justifications for their methods and the limitations of their study. Their responses demonstrate careful consideration of my review comments and a satisfactory effort to improve the manuscript's clarity, robustness, and generalizability.

There are areas, particularly in “discussing the impact of excluding rarer diseases” and “further exploring the generalizability of the findings,” where the revision could provide even greater depth and promise for the study's broader applicability.

Overall, the revisions and responses indicate significant improvements to the manuscript. Therefore, I am pleased to accept the article as is or after addressing my minor comment above. I leave this decision to the editor.

Response: We would like to express our gratitude to the reviewer for their time and effort. We have made revisions to the discussions as per the suggestions (lines 477-482, Methods section).

Reviewer #2 (Remarks to the Author):

Thank you for your thorough response to my comments.

Response: Thank you for taking the time to provide valuable feedback on our work.